# Interfacial piezoelectric polarization locking in printable Ti₃C₂Tₓ MXene-fluoropolymer composites

Nick A. Shepelin [1,2,8], Peter C. Sherrell [1,2], Emmanuel N. Skountzos[3,4], Eirini Goudeli[1], Jizhen Zhang [5], Vanessa C. Lussini[6], Beenish Imtiaz[1], Ken Aldren S. Usman[5], Greg W. Dicinoski[6], Joseph G. Shapter[7], Joselito M. Razal[5] & Amanda V. Ellis [1,2 ✉]

Piezoelectric fluoropolymers convert mechanical energy to electricity and are ideal for sustainably providing power to electronic devices. To convert mechanical energy, a net polarization must be induced in the fluoropolymer, which is currently achieved via an energy-intensive electrical poling process. Eliminating this process will enable the low-energy production of efficient energy harvesters. Here, by combining molecular dynamics simulations, piezoresponse force microscopy, and electrodynamic measurements, we reveal a hitherto unseen polarization locking phenomena of poly(vinylidene fluoride–co–trifluoroethylene) (PVDF-TrFE) perpendicular to the basal plane of two-dimensional (2D) Ti₃C₂Tₓ MXene nanosheets. This polarization locking, driven by strong electrostatic interactions enabled exceptional energy harvesting performance, with a measured piezoelectric charge coefficient, $d_{33}$, of −52.0 picocoulombs per newton, significantly higher than electrically poled PVDF-TrFE (approximately −38 picocoulombs per newton). This study provides a new fundamental and low-energy input mechanism of poling fluoropolymers, which enables new levels of performance in electromechanical technologies.

[1] Department of Chemical Engineering, The University of Melbourne, Parkville, VIC, Australia. [2] BioFab3D, Aikenhead Centre for Medical Discovery, St Vincent's Hospital Melbourne, Fitzroy, VIC, Australia. [3] Department of Chemical Engineering, University of Patras, Patras, Greece. [4] FORTH/ICE-HT, Patras, GR, Greece. [5] Institute for Frontier Materials, Deakin University, Geelong, VIC, Australia. [6] Note Issue Department, Reserve Bank of Australia, Craigieburn, VIC, Australia. [7] Australian Institute for Bioengineering and Nanotechnology, The University of Queensland, Brisbane, QLD, Australia. [8]Present address: Laboratory for Multiscale Materials Experiments, Paul Scherrer Institut, Villigen, Switzerland. ✉email: amanda.ellis@unimelb.edu.au

For dielectric materials exhibiting piezoelectricity, inducing polarization through the alignment of the dipoles is paramount to couple mechanical and electrical energy[1]. To achieve dipole alignment, electrical poling is considered a necessary task in the post-processing of piezoelectric materials (Fig. 1a–c)[2]. Electrical poling is energy-intensive, with electric fields on the order of tens to hundreds of megavolts per meter commonly used (Fig. 1b, c)[3–5]. In fluoropolymers such as poly (vinylidene fluoride) (PVDF), a class of semicrystalline linear-chain polymers exhibiting a dipole moment between the hydrogen and fluorine moieties perpendicular to the carbon backbone (Fig. 1a), the poling process additionally requires elevated temperature conditions[3,4,6]. The highly valorized commercial applications for piezoelectric materials, including precision motorized stages and inkjet printheads, utilize the converse piezoelectric effect[7], converting an applied electric field to discrete mechanical outputs[6]. In contrast, emerging applications that utilize the direct piezoelectric effect[7] to convert mechanical to electrical energy, the electrical poling process is a roadblock to commercialization, requiring a higher energy input than can be harvested in the device lifespan. These emergent applications, including energy harvesting[3,8], robotic interfaces[9], piezocatalysis[10], and piezophotonics[11], require revisiting of the electrical poling process and examination of the pathways for inducing polarization without high-input energies[1,12].

Recent efforts have investigated alternative pathways to polarize fluoropolymers by tuning solvent composition[13] or using nanofillers[8,14–18]. The templating of polarization has been realized by nanomaterial–fluoropolymer interactions from piezoelectric BaTiO₃ nanoparticles[14,15], reduced graphene oxide nanosheets[16,17], hexagonal boron nitride nanoflakes[18], and single-walled carbon nanotubes[8]. However, the mechanism of

dipole alignment arising from templating piezoelectric polymers with nanofillers remains poorly understood. Notably, the aforementioned nanomaterials provide limited scope to probe dipole moment alignment, as they possess piezoelectric properties such as $BaTiO_3$[14], or they alter the number of dipoles through changes in polymer conformation (e.g., by reduced graphene oxide or hexagonal boron nitride)[16,17]. The single-walled carbon nanotube template has shown promise on a nanoscale[8], although the mechanism of polarization templating was not elucidated.

To investigate the mechanism of polarization templating (Fig. 1d), a nanofiller must have out-of-plane polarizability without out-of-plane piezoelectric properties[19]. Further, it should be mechanically rigid with well-defined surface properties and functionality. To this end, the rapidly evolving class of transition metal carbides (MXenes) is an excellent candidate to probe polarization templating, with $Ti_3C_2T_x$ being the most well-characterized MXene[20–27]. Importantly, it has out-of-plane polarizability[21] with symmetry perpendicular to the basal plane[23,28] and is therefore hypothesized to not possess out-of-plane piezoelectric properties.

In this work, we provide a deep mechanistic understanding of how polarization templating can be achieved in fluoropolymers from $Ti_3C_2T_x$ nanosheet templates (Fig. 1d, e). We employ molecular dynamics (MD) simulations to probe the evolution of the polarization of PVDF-TrFE in relation to the $Ti_3C_2T_x$ nanosheet, revealing that the electrostatic interactions between the $Ti_3C_2T_x$ nanosheet and the fluoropolymer are crucial to achieve effective induced local polarization locking. We then extend this induced local polarization locking to a macroscale net polarization using solvent-evaporation assisted (SEA) 3D printing to impart enhanced shear alignment[8,29]. The resultant

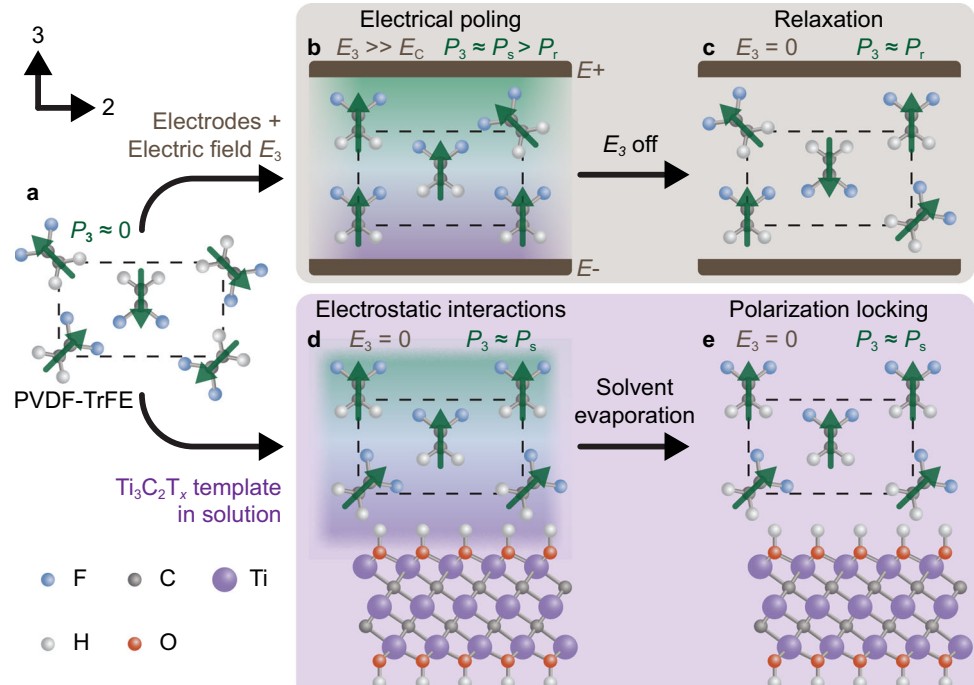

**Fig. 1 Simplified schematic outlining the nanomaterial-induced polarization locking mechanism in PVDF-TrFE as an alternative to the conventional electrical poling method. a** The β phase PVDF-TrFE chains, obtained directly following film deposition, exhibit a randomized dipolar orientation (green arrows), resulting in a negligible net polarization ($P$). **b** In the electrical poling method, electrodes are attached to the surfaces perpendicular to the desired polarization direction and an electric field ($E$) is applied, significantly higher than the coercive field ($E_C$) in order to orient the individual dipole moment vectors and maximize the $P$ to the spontaneous polarization ($P_s$). **c** Following the removal of the $E$, the PVDF-TrFE chains undergo partial relaxation from $P_s$ to the remnant polarization ($P_r$). **d** Conversely, adding the $Ti_3C_2T_x$ ($T_x \approx OH$) nanosheets to the PVDF-TrFE in the solution enables the $P$ to align, without an applied electric field, perpendicular to the basal plane of the $Ti_3C_2T_x$ nanosheets via electrostatic interactions at the interface. **e** Following deposition, the $Ti_3C_2T_x$ nanosheets are generally aligned parallel to the substrate, and the subsequent evaporation of the solvent locks the $P$ at $P_s$ with no relaxation.

composites show a piezoelectric charge coefficient ($d_{33}$), voltage coefficient ($g_{33}$), and figure of merit of $-52.0$ pC N$^{-1}$, 402 mV m N$^{-1}$, and $20.9 \times 10^{-12}$ Pa$^{-1}$, respectively, higher than electrically poled PVDF-TrFE co-polymer ($\sim -38$ pC N$^{-1}$, 380 mV m N$^{-1}$, and $14.4 \times 10^{-12}$ Pa$^{-1}$, respectively)[3,30], which demonstrates that our composites are fully polarized within the electroactive crystalline phase. The advancements herein enable rapid, cost-effective, energy-efficient, and scalable production of fluoropolymers for emerging applications utilizing the direct piezoelectric effect, including as a power supply for broad-scale wearable electronics.

## Results

**Preparation of Ti$_3$C$_2$T$_x$/PVDF-TrFE composite SEA 3D printing inks**. Ti$_3$C$_2$T$_x$ nanosheets are a 2D material with the point group of $P6_3/mmc$, which is symmetric (and therefore non-piezoelectric) in the z-direction (i.e., no out-of-plane polarization)[23,28]. The Ti$_3$C$_2$T$_x$ nanosheets were exfoliated from a Ti$_3$AlC$_2$ MAX phase (Supplementary Fig. 1a) using the minimally intensive layer delamination exfoliation method with LiF and HCl, exhibiting an average lateral size of 310 nm (Supplementary Fig. 1b, c) and a thickness of $\sim$1 nm (Supplementary Fig. 1d)[24,27]. This soft exfoliation technique resulted in a dominant OH surface termination (T$_x$), with trace amounts of F and O functionality (Supplementary Information Fig. 1e–g)[25,26].

Composites of Ti$_3$C$_2$T$_x$ nanosheets and PVDF-TrFE were prepared as inks by a simple mixing process, whereby a small volume of the Ti$_3$C$_2$T$_x$ nanosheets in N,N-dimethylformamide (DMF) was added to a PVDF-TrFE (40 wt%) solution in acetone and homogenized via stirring at room temperature. Concentrations between 0.00 wt% and 0.50 wt% Ti$_3$C$_2$T$_x$, relative to the mass of PVDF-TrFE, were produced as viscous inks (Supplementary Fig. 2). Interestingly, all inks showed sustained stability, with retention of their initial color and flow properties for up to 5 months post mixing (Supplementary Fig. 3). This stability is primarily due to the retardation of either/or the surface oxidation and agglomeration of the Ti$_3$C$_2$T$_x$ nanosheets that can typically occur[26].

**MD simulations of the Ti$_3$C$_2$T$_x$/PVDF-TrFE interface**. To understand the interface between the Ti$_3$C$_2$T$_x$ nanosheets and the PVDF-TrFE co-polymer, MD calculations were performed using the periodic lattice of Ti$_3$C$_2$T$_x$ (where T$_x$ = OH) and 70 "mer" chains of PVDF-TrFE (Fig. 2a, left). This was then compared with a periodic lattice of graphene (Fig. 2b, left) with an equivalent polymer film. These simulations revealed an extremely strong electrostatic interaction between the PVDF-TrFE chains and the Ti$_3$C$_2$T$_x$ nanosheet, requiring $\sim$4.17 pN of force to desorb one PVDF-TrFE chain from the Ti$_3$C$_2$T$_x$. This strong interaction limits the motion of the polymer (Supplementary Movie 1), with chains adjacent to the surface elongated and unable to move, forming a tightly packed structure (Fig. 2a) with a local density of $\sim$1.6 g cm$^{-3}$ (Supplementary Fig. 5). Over 4 ns, more chains preferentially fill the free space on the Ti$_3$C$_2$T$_x$ lattice, leading to the decrease in the density of the second layer. Interestingly, there is no statistical difference between the H and F positions on the PVDF-TrFE chains relative to the Ti$_3$C$_2$T$_x$ nanosheet (Supplementary Fig. 6, shown for a film of 14 chains). There is a clear difference in the proportion of *trans* (63%) and *gauche* (37%) conformations within the PVDF-TrFE chains at the interface of the Ti$_3$C$_2$T$_x$ nanosheet (Supplementary Figs. 7 and 8). This ratio of bond conformations suggests the inhibition of local electroactive phase within the fluoropolymer.

The orientation of the net dipole moment vector of the PVDF-TrFE co-polymer film was compared with that of the Ti$_3$C$_2$T$_x$ nanosheet, quantified through the angle $\theta$ (Fig. 2a, right). The spatial and temporal evolution of $\theta$ against the separation ($s$) reveals the fluoropolymer layers adjacent to the Ti$_3$C$_2$T$_x$ nanosheet (within 5 Å from the Ti$_3$C$_2$T$_x$ nanosheet surface, directly experiencing the strong electrostatic attraction) have a preferential orientation parallel to the Ti$_3$C$_2$T$_x$ nanosheet (Fig. 2c). However, with increasing separation from the Ti$_3$C$_2$T$_x$ nanosheet, this PVDF-TrFE co-polymer net dipole moment becomes increasingly perpendicular to the basal plane of the Ti$_3$C$_2$T$_x$ nanosheet substrate. Interestingly, thicker co-polymer films (within 39 Å from the Ti$_3$C$_2$T$_x$ nanosheet surface) exhibit an extremely tight distribution of the polarization orientation (Fig. 2c), which remains perpendicular to the Ti$_3$C$_2$T$_x$ nanosheet for the entire simulation (Supplementary Fig. 9, Supplementary Movie 1). These results show that on a local scale, a net polarization of the PVDF-TrFE co-polymer is formed perpendicular to the basal plane of the Ti$_3$C$_2$T$_x$ nanosheet (Supplementary Movie 2). Given that the electrostatic screening length in 2D materials is between 1 nm and 10 nm[31], and considering the strength of the electrostatic interaction observed between the PVDF-TrFE co-polymer and the Ti$_3$C$_2$T$_x$ nanosheet, it is highly probable that this polarization locking is occurring as a consequence of electrostatic forces[19].

For comparison, the net dipole moment vector of the same PVDF-TrFE co-polymer film on a graphene sheet was investigated under identical simulation conditions (Fig. 2b, right). As an atomically thin 2D sheet, graphene does not exhibit out-of-plane polarizability and is thus a suitable comparative substrate system to Ti$_3$C$_2$T$_x$[19]. In contrast to the fluoropolymer film on Ti$_3$C$_2$T$_x$ nanosheet, the fluoropolymer film on graphene is able to migrate on the periodic lattice easily (Supplementary Movie 1) and requires a significantly lower force ($\sim$2.78 pN) to detach a single fluoropolymer chain from the lattice, indicating a weaker interaction between the components. The polarization vector of the fluoropolymer film always exhibits random orientations with respect to the graphene surface, regardless of the layer thickness (Fig. 2b, right, Fig. 2d, Supplementary Movie 1), and this phenomenon is observed for individual PVDF-TrFE co-polymer chains (Supplementary Movie 2).

The MD simulations show a clear and strong binding interaction between the Ti$_3$C$_2$T$_x$ nanosheet and the PVDF-TrFE co-polymer chains, driven by electrostatic interactions. This binding results in the subsequent self-assembly of the fluoropolymer film, oriented in such a way as to tightly lock the polarization of the PVDF-TrFE perpendicular to the Ti$_3$C$_2$T$_x$ basal plane, providing guidance towards an experimental tool for self-assembly driven polarization in bulk fluoropolymer materials.

**Towards printing and net polarization**. For the translation of this fundamental understanding of the induced local net polarization, developed using MD simulations, into macroscale piezoelectric generators (PEGs), a careful selection of the Ti$_3$C$_2$T$_x$/PVDF-TrFE composite processing route is required. SEA 3D printing can enable shear alignment of both nanofillers[27,29] and polymer chains[8,32], parallel to the direction of printing. Given the formation of induced local polarization, perpendicular to the basal plane of the Ti$_3$C$_2$T$_x$ nanosheets, this parallel alignment is hypothesized to give out-of-plane piezoelectricity without the need for electric poling.

To understand the interaction between the Ti$_3$C$_2$T$_x$ nanosheets and the PVDF-TrFE co-polymer in solution, as well as assess the suitability for relevant solution processing routes including extrusion printing, the shear strain ($\gamma_s$) response of the Ti$_3$C$_2$T$_x$ loaded PVDF-TrFE (40 wt%) inks was studied (Fig. 3a). The extended rheological characterization of the pristine PVDF-TrFE

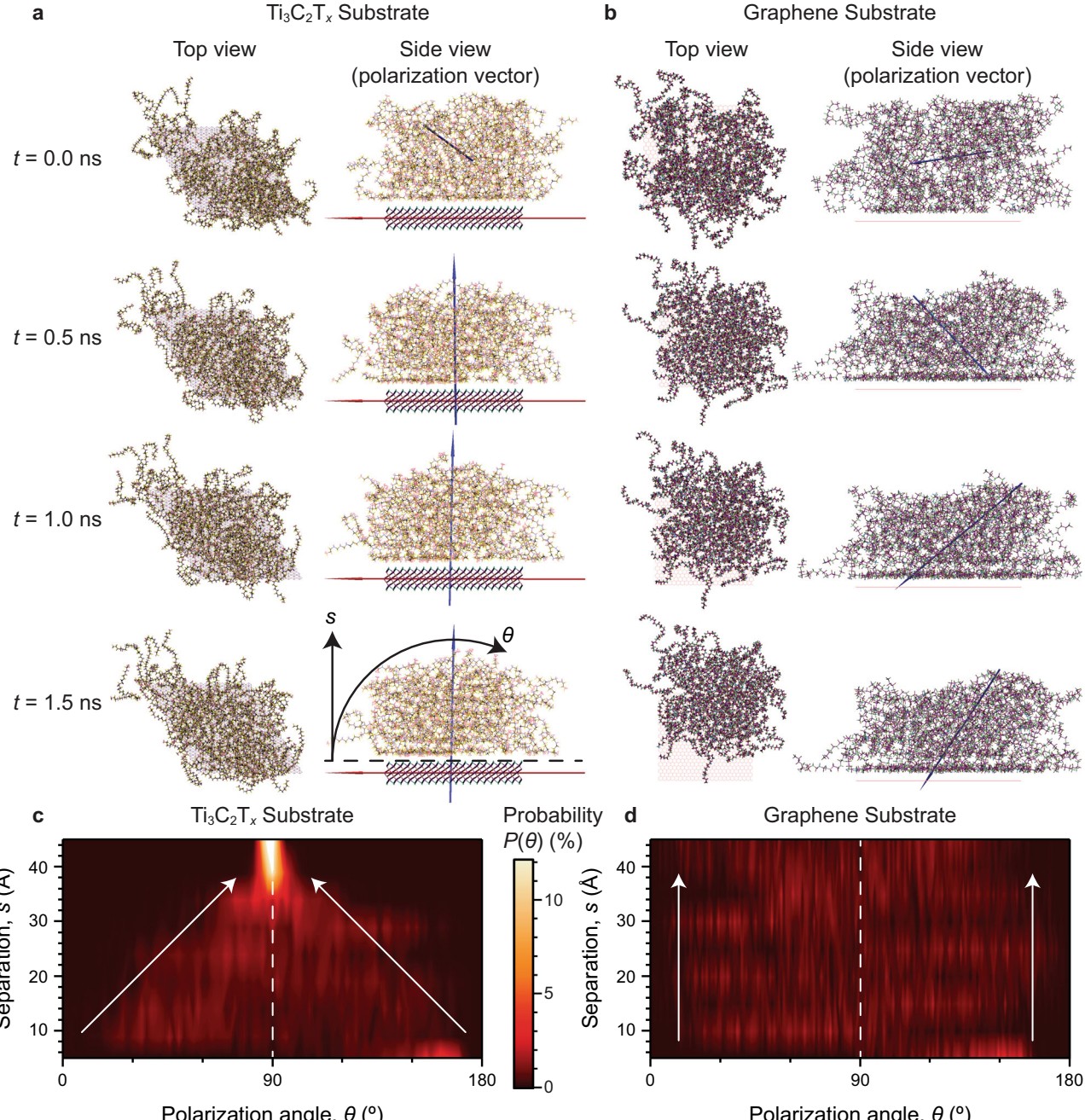

**Fig. 2 Comparative MD simulations of the PVDF-TrFE co-polymer film polarization on Ti$_3$C$_2$T$_x$ and graphene substrates. a**, **b** The top view (left column) and side view (right column) snapshots at $t = 0$, 0.5, 1, and 1.5 ns, with the resultant polarization vectors (right column) of the PVDF-TrFE co-polymer film (blue arrow) and **a** Ti$_3$C$_2$T$_x$ substrate (red arrow) or **b** graphene substrate. The annotated $s$ and $\theta$ in **a** represent the separation of the fluoropolymer from the substrate and the PVDF-TrFE co-polymer polarization angle relative to the substrate, respectively. **c**, **d** Probability distributions of the angle ($\theta$) between the polarization vector of the PVDF-TrFE co-polymer film layers and the **c** Ti$_3$C$_2$T$_x$ or **d** graphene substrates as a function of separation ($s$), as calculated from the equilibrated region of the obtained MD simulation.

ink is presented in the Supplementary Information (Supplementary Figs. 10–12). Both the pristine PVDF-TrFE co-polymer and the Ti$_3$C$_2$T$_x$/PVDF-TrFE inks demonstrated exceptional flow properties for printing. This was clearly demonstrated by the storage modulus ($G'$) being greater than the loss modulus ($G''$) at low $\gamma_s$, indicating the ability of the ink to retain a physical shape. With increasing $\gamma_s$, analogous to the strain applied during extrusion printing, both inks exhibited yielding and liquid-like behavior, indicated by the cross-over of $G'$ and $G''$, and the subsequent region with $G' < G''$. Similarly, the angular frequency

($\omega$) response (Fig. 3b) confirmed the formation of a strong physical gel in both the pristine PVDF-TrFE co-polymer and the Ti$_3$C$_2$T$_x$/PVDF-TrFE inks, highly desirable for extrusion printing[33]. A weakening of the PVDF-TrFE/acetone interaction was apparent through the presence of a larger shoulder at $\omega \approx$ 0.15 rad s$^{-1}$ in the $G''$ of the Ti$_3$C$_2$T$_x$/PVDF-TrFE ink. The weakening of the physical gel was attributed to a strong interaction between the PVDF-TrFE co-polymer and the Ti$_3$C$_2$T$_x$ nanosheets. However, the Ti$_3$C$_2$T$_x$/PVDF-TrFE ink, nonetheless, exhibited solid-like behavior ($G' > G''$) at $\omega$ as low as 0.10 rad s$^{-1}$.

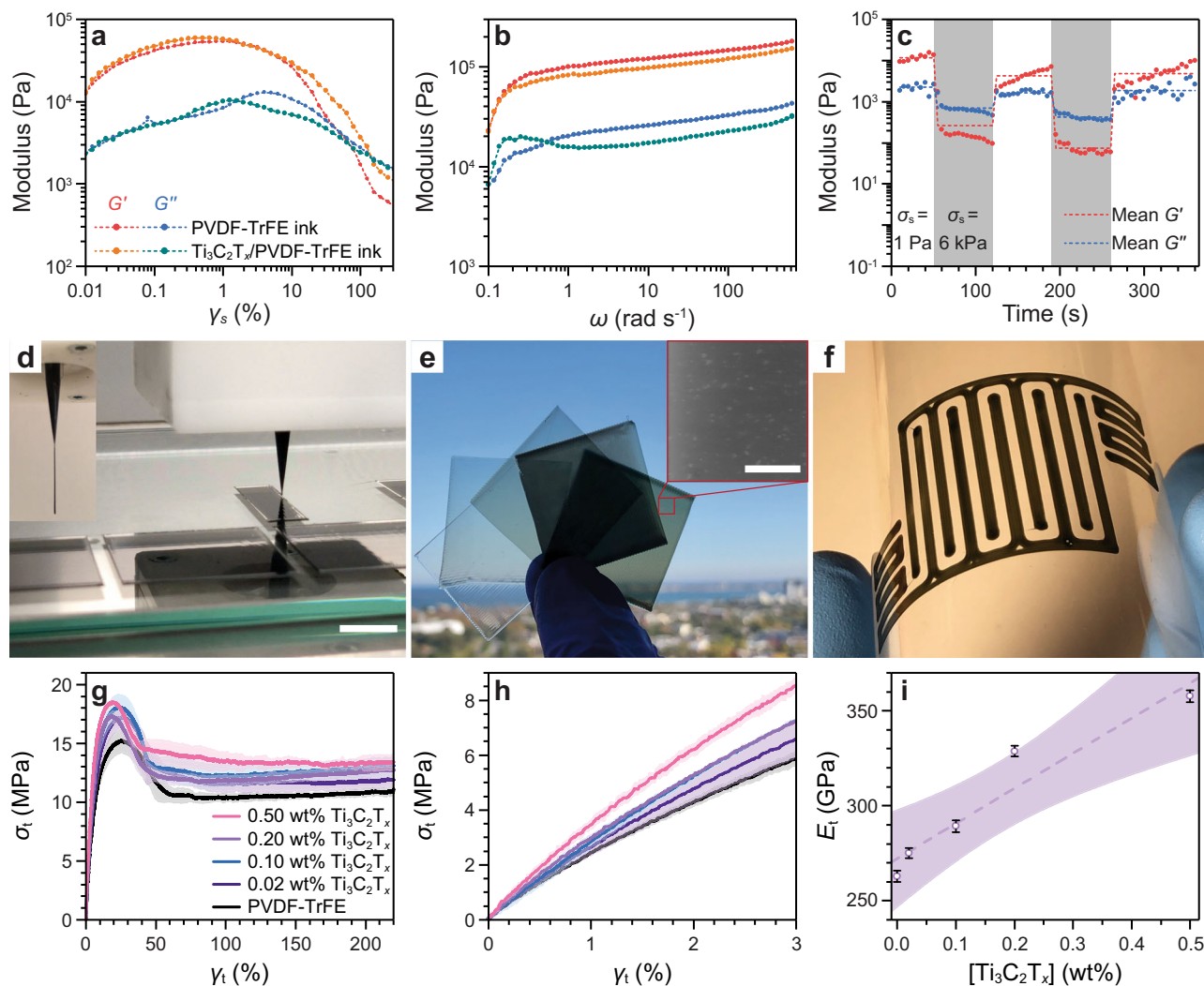

**Fig. 3 SEA extrusion printing of strengthened Ti₃C₂Tₓ/PVDF-TrFE films. a** Oscillatory shear strain ($\gamma_s$) sweep ($\omega = 1\,Hz$) and **b** oscillatory frequency ($\omega$) sweep ($\gamma_s = 1\%$) of the 40 wt% PVDF-TrFE ink with and without 0.20 wt% Ti₃C₂Tₓ nanosheets. **c** The temporal evolution of the storage ($G'$) and loss ($G''$) moduli during oscillatory shear stress ($\sigma_s$) cycling between 1 Pa (white regions, representing the stationary state) and 6 kPa (gray regions, representing printing stress). **d** Photograph showing the SEA extrusion printing process for the Ti₃C₂Tₓ/PVDF-TrFE (0.50 wt%) ink, scale bar 10 mm. Inset shows a self-supporting single filament. **e** Photograph of the free-standing Ti₃C₂Tₓ/PVDF-TrFE films at various Ti₃C₂Tₓ nanosheet loadings (from left to right: 0.00 wt%, 0.02 wt%, 0.10 wt%, 0.20 wt%, 0.50 wt%). Inset shows helium ion beam microscopy (HIM) image of Ti₃C₂Tₓ/PVDF-TrFE (0.50 wt%) film surface, demonstrating the distribution of Ti₃C₂Tₓ nanosheets near the surface of the film. Scale bar represents 40 μm. **f** Photograph demonstrating an interdigitated structure of 0.50 wt% Ti₃C₂Tₓ/PVDF-TrFE composite printed onto a flexible poly(ethylene terephthalate) (PET) substrate. **g** Tensile strain ($\gamma_t$)-stress ($\sigma_t$) profiles of the printed Ti₃C₂Tₓ/PVDF-TrFE films (shaded region shows the error from replicates). **h** The elastic region of **g**. **i** The Young's modulus ($E_t$) obtained from **h**. The overlay in **g** and **h**, and the error bars in **i** represent the mean ±SD. The overlay in **i** represents a 95% CI for a linear fit.

The PVDF-TrFE co-polymer ink was subjected to consecutive small amplitude oscillatory shear (SAOS, 1 Pa) and large amplitude oscillatory shear (5 kPa) cycling at 1 Hz to replicate the shear stress of extrusion printing, shown in Fig. 3c[34]. It was found to consistently flow under 5 kPa shear stress ($\sigma_s$), with immediate $G''$ recovery and >60% $G'$ recovery after 70 s at $\sigma_s = 1$ Pa, retaining similar characteristics with >85% G' recovery on the second $\sigma_s$ cycle. This testing method showed the ability of the PVDF-TrFE co-polymer and Ti₃C₂Tₓ/PVDF-TrFE inks to print continuous and complex structures without void formation or defects in the resultant structures.

**SEA extrusion printed and mechanically robust Ti₃C₂Tₓ/PVDF-TrFE composite films.** With a clear understanding of the rheological performance of the Ti₃C₂Tₓ/PVDF-TrFE inks, as well as evidence of weakening of the PVDF-TrFE/acetone interaction

owing to the strong Ti₃C₂Tₓ/PVDF-TrFE interface, thin films were prepared for studies of the macroscale polarization and energy harvesting properties.

Ti₃C₂Tₓ/PVDF-TrFE inks at various Ti₃C₂Tₓ nanosheet loadings (0.00 wt%, 0.02 wt%, 0.10 wt%, 0.20 wt%, 0.50 wt%) were SEA extrusion printed as single-layer thin films (Fig. 3d) through a nozzle (internal diameter = 200 μm) onto a clean glass plate and then subsequently removed to form free-standing films (Fig. 3e). The distribution of the Ti₃C₂Tₓ nanosheets was homogeneous in the resultant films with the basal plane appearing aligned parallel to the printing direction (Fig. 3e, inset), as expected from an extrusion process. This SEA extrusion printing process could be further extended to print complex shapes on flexible substrates, such as poly(ethylene terephthalate), as shown in Fig. 3f. The versatility of the SEA extrusion printing technique enables the deposition of multi-layer systems[32], on conformal[35] and moving

substrates[36], allowing our PEG to be deployed in broad and highly specialized applications such as in the point-of-care printing of in vivo energy harvesters[37,38].

The mechanical properties of the SEA extrusion printed $Ti_3C_2T_x$/PVDF-TrFE (0.00 wt%, 0.02 wt%, 0.10 wt%, 0.20 wt%, 0.50 wt%) composite films were studied via tensile extension (Fig. 3g). All films were shown to yield at tensile strain ($\gamma_t$) <50% and did not break at $\gamma_t = 220\%$ (displacement limit of the instrument), confirming the high ductility of all tested materials. Subsequent elongation at uncontrolled force by hand showed a maximum $\gamma_t > 1000\%$ prior to failure (Supplementary Fig. 13). The Young's modulus ($E_t$), obtained from the low-$\gamma_t$ region (Fig. 3h), increased linearly with an increase in $Ti_3C_2T_x$ nanosheet loadings in the PVDF-TrFE co-polymer up to 358 MPa at a $Ti_3C_2T_x$ nanosheet loading of 0.5 wt% (Fig. 3i). This linear increase showed that there is a homogenous dispersion of the $Ti_3C_2T_x$ nanosheets through the fluoropolymer at all concentrations, as aggregation would lead to poor load transfer between the components. The $E_t$ of the pristine PVDF-TrFE co-polymer film (263 MPa) was comparable to literature values for 3D printed PVDF (419 MPa), although lower than other processing routes[39].

**PVDF-TrFE phase dependence in $Ti_3C_2T_x$/PVDF-TrFE composites.** The semicrystalline PVDF-TrFE co-polymer exists in three favorable conformations, the symmetric and therefore non-electroactive α phase, the semi-polar and moderately electroactive γ phase, and the polar and highly electroactive β phase[3]. These phases arise from the presence and order of *trans* (T) and *gauche* (G) bond conformations, whereby the α phase consists of alternating conformations (TGTG'), the γ phase is an intermediate (TTTG) and the β phase is all-*trans* (TTTT)[4]. Inherently, changes in the crystallinity and phase composition affect the maximum polarization of fluoropolymers. Understanding how these parameters change is crucial to elucidating a mechanism for an enhanced piezoelectric response. Ultimately, entropy within the polymer during deposition results in negligible polarization of the polymer films, thus requiring electrical poling to align the dipole moment vectors[8]. To achieve this understanding of crystallinity and phase composition, thorough phase characterization (differential scanning calorimetry (DSC) and Raman confocal microscopy) was performed (Fig. 4). These tools enable the determination of both the total crystallinity of the fluoropolymer and the relative proportion of β and γ phases, enabling the exhaustive understanding of the material for the assessment of energy harvesting capabilities.

Raman spectroscopy showed a clear increase in the $Ti_3C_2T_x$ nanosheet spectrum vibrational modes (150 $cm^{-1}$–740 $cm^{-1}$) with increasing concentration (Fig. 4a). An initial decrease was observed in the intensity ratio $I_\beta/I_\gamma$ (β phase at 840 $cm^{-1}$ and γ phase at 811 $cm^{-1}$) from 2.4 for the pristine PVDF-TrFE co-polymer to 2.2 for the 0.02 wt% $Ti_3C_2T_x$/PVDF-TrFE composite, before an increase to 2.5 for the 0.5 wt% $Ti_3C_2T_x$/PVDF-TrFE composite (Fig. 4b)[3].

DSC revealed that small $Ti_3C_2T_x$ nanosheet loadings (0.02 wt%) resulted in a notable decrease in the overall crystallinity of the PVDF-TrFE co-polymer matrix (Fig. 4c).

Confocal Raman mapping was also performed (Fig. 4d, e) to probe the decrease in both the $I_\beta/I_\gamma$ and the crystallinity, as shown by the average Raman spectra (Fig. 4b) and DSC (Fig. 4c), respectively. Notably, the analysis of these maps found a clear difference in the $I_\beta/I_\gamma$ between the bulk PVDF-TrFE co-polymer (Fig. 4d, black circle) and the $Ti_3C_2T_x$/PVDF-TrFE interface (Fig. 4d, red circle), decreasing from 2.65 to 1.45, respectively. The individual spectra for these regions (Fig. 4e) showed a

disproportionate suppression of the β phase peak intensity at the $Ti_3C_2T_x$ nanosheet surface, whereas the intensity ratio between γ and the $CH_2$ stretch (1432 $cm^{-1}$) maintains the same ratio[40]. Importantly, these findings suggest a local inhibition of the electroactive β phase crystallization at the interface between the $Ti_3C_2T_x$ nanosheet and the PVDF-TrFE co-polymer (Supplementary Fig. 7), as well as the local densification of the fluoropolymer film (Supplementary Fig. 5).

Thus, the decrease in the crystallinity at 0.02 wt% $Ti_3C_2T_x$/PVDF-TrFE (Fig. 4c) was attributed to a lower fraction of the electroactive β phase in the PVDF-TrFE co-polymer forming directly on the $Ti_3C_2T_x$ nanosheet surface.

Given the understanding developed through MD simulations (Fig. 2), the fluoropolymer densification was hypothesized to occur in solution prior to printing, rather than during printing. There are two factors to consider here: (1) the effect of shear alignment on the fluoropolymer itself[32,41], and (2) the effect of shear on orienting the $Ti_3C_2T_x$ nanosheets[27,29]. At 0.02 wt% $Ti_3C_2T_x$ nanosheets, it is proposed that the decrease in the β phase at the PVDF-TrFE/$Ti_3C_2T_x$ interface, coupled to the minimal shear orientation of the $Ti_3C_2T_x$ nanosheet, has a stronger negative effect compared with shear aligning of the fluoropolymer molecules. For higher $Ti_3C_2T_x$ nanosheet loadings, extruded through the nozzle, a greater shear alignment phenomenon is observed within the material, resulting in a very slight net increase in β phase and overall crystallinity. This hypothesis is supported by the Raman spectra of the solvent cast 0.50 wt% $Ti_3C_2T_x$/PVDF-TrFE composite film (Fig. 4b, Supplementary Fig. 15f), which does not undergo shear-induced crystallization and has a $I_\beta/I_\gamma$ comparable to the printed 0.02 wt% $Ti_3C_2T_x$/PVDF-TrFE composite film.

**Energy harvesting of the $Ti_3C_2T_x$/PVDF-TrFE composite films.** Prior to printing, the strong electrostatic binding interactions between the $Ti_3C_2T_x$ nanosheets and the PVDF-TrFE co-polymer chains, as described by the MD simulations, enable the $Ti_3C_2T_x$ nanosheet to remain sterically stabilized in the inks, without aggregation (Supplementary Fig. 3). This lack of $Ti_3C_2T_x$ nanosheet aggregation enables the enhancement in the $E_t$ (Fig. 3i). To understand how this induced local polarization locking translates to bulk and macroscopic energy harvesting, both piezoelectric force microscopy (PFM) and bulk electromechanical testing were performed.

PFM was carried out by applying a bias between −20 V and +20 V to a conductive platinum (Pt) cantilever in contact with the $Ti_3C_2T_x$/PVDF-TrFE films (at various $Ti_3C_2T_x$ nanosheet loadings) and the subsequent local changes in thickness (arising from the expansion or contraction of the unit cell in the polarized electroactive phases of the PVDF-TrFE co-polymer) were measured[6]. At voltages below the poling electric field (<50 MV $m^{-1}$), piezoelectric materials exhibit a strong correlation between the induced strain ($\gamma_3$) and the $d_{33}$[3,4]. In PFM, the correlation is qualitative[42] and the $d_{33} = A\cos(\varphi)/Q_fV$, where $A$ is the amplitude, $\varphi$ is the phase, $Q_f$ is the Q-factor of the cantilever, and $V$ is the applied bias[8]. The extended discussion surrounding the PFM is presented in the Supplementary Information.

To confirm the lack of piezoelectric contribution arising in the z-direction of the $P6_3/mmc$ point group of the $Ti_3C_2T_x$, a $Ti_3C_2T_x$ nanosheet on gold-coated silicon (Au@Si) substrate was probed using dual AC resonance tracking PFM, whereby a constant bias (1 V) was applied to the cantilever (Supplementary Fig. 20). The $Ti_3C_2T_x$ nanosheet, ~300 nm in the lateral dimension, was visible topographically, however, no discernible changes were observed in the $A$ and $\varphi$ traces between the $Ti_3C_2T_x$ nanosheet and the underlying substrate (Fig. 5a). Therefore, the $Ti_3C_2T_x$ nanosheet

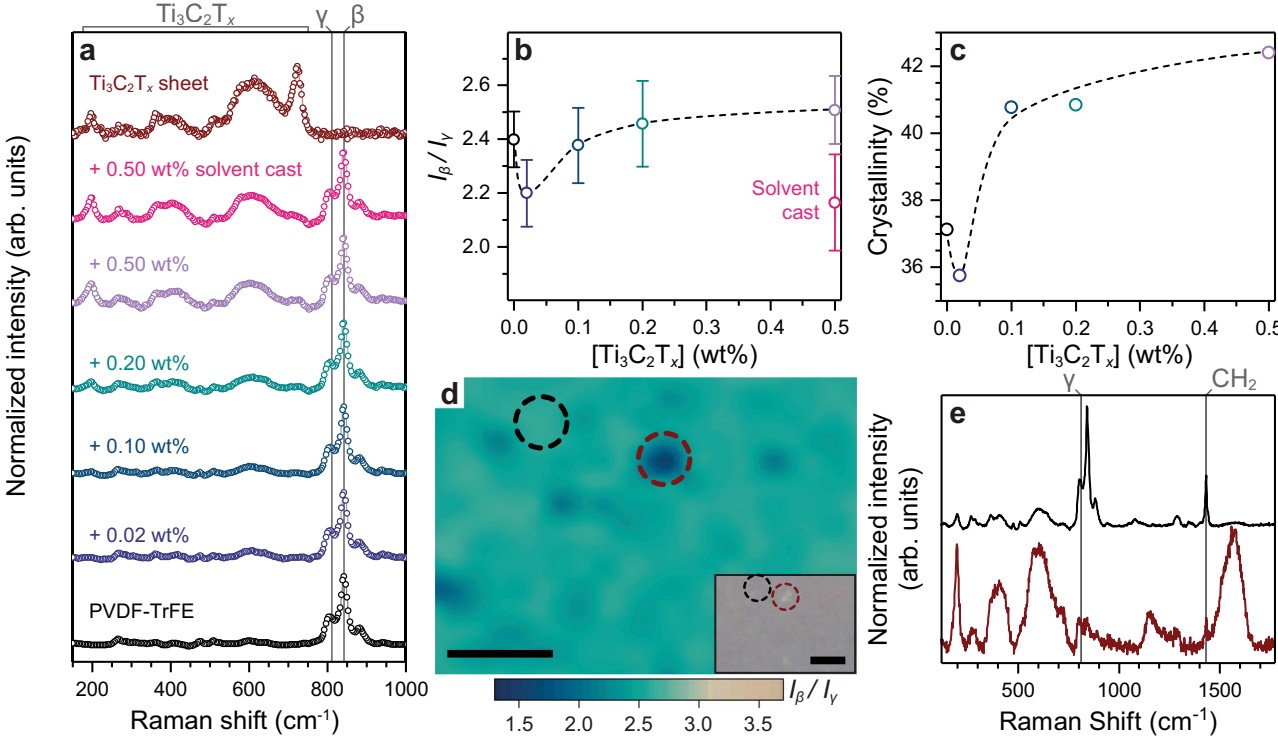

**Fig. 4 Material characteristics of Ti$_3$C$_2$T$_x$/PVDF-TrFE films. a** Raman spectra of Ti$_3$C$_2$T$_x$/PVDF-TrFE composites (at various Ti$_3$C$_2$T$_x$ loadings) and Ti$_3$C$_2$T$_x$ nanosheets. **b** The β/γ phase ratio, $I_\beta/I_\gamma$, calculated from the intensities of the β phase (840 cm$^{-1}$) and γ phase (811 cm$^{-1}$) from Raman confocal mapping (Supplementary Fig. 15). The error bars represent the mean ±SD in $I_\beta/I_\gamma$ over each Raman map. **c** The effect of Ti$_3$C$_2$T$_x$ nanosheet loading on the PVDF-TrFE co-polymer crystallinity, as measured by DSC. The dashed lines in **b** and **c** are visual guides. **d** The Raman $I_\beta/I_\gamma$ map of the Ti$_3$C$_2$T$_x$/PVDF-TrFE (0.50 wt%) film, where the dark point designated by the red circle corresponds to a large Ti$_3$C$_2$T$_x$ nanosheet at the film surface. The inset shows a confocal microscope image of the area analyzed using Raman confocal mapping. Both scale bars represent 5 μm. **e** The representative Raman spectra for the area designated by the black circle and the red circle in **d**, showing the clear inhibition of β phase locally at the Ti$_3$C$_2$T$_x$ nanosheet surface.

exhibited no observable out-of-plane piezoelectric effect (perpendicular to the nanosheet basal plane).

To measure the $d_{33}$ of the composite Ti$_3$C$_2$T$_x$/PVDF-TrFE films, a variable bias was applied, with the spatial map of the applied bias shown in Supplementary Fig. 21. The representative PFM response for the SEA extrusion printed Ti$_3$C$_2$T$_x$/PVDF-TrFE (0.00 wt%, 0.02 wt%, 0.10 wt%, 0.20 wt%, and 0.50 wt%) films is shown in Fig. 5b (extended data presented in Supplementary Figs. 22 and 23). The amplitude, corrected for the directionality of strain and the Q-factor of the cantilever ($A\cos(\varphi)/Q_f$), exhibited a maximum at −41.6 pm for the pristine PVDF-TrFE co-polymer film at −20 V (Supplementary Fig. 24), confirming a spatial alignment effect arising from shear stresses during the extrusion printing process, as previously reported[8,32]. The $A\cos(\varphi)/Q_f$ value was shown to increase sharply with an increase in Ti$_3$C$_2$T$_x$ nanosheet loading within the PVDF-TrFE co-polymer (Supplementary Fig. 24d). A substantial increase was observed in the maximum $A\cos(\varphi)/Q_f$ at −20 V for the Ti$_3$C$_2$T$_x$/PVDF-TrFE (0.50 wt%) SEA extrusion printed film (−182.3 pm), compared to the pristine PVDF-TrFE co-polymer film (−41.6 pm), indicating an intensified piezoelectric response upon the addition of the Ti$_3$C$_2$T$_x$ nanosheets.

The measured $d_{33}$ as a function of Ti$_3$C$_2$T$_x$ nanosheet concentration in the printed films is shown in Fig. 5c. The effective $d_{33}$ for the pristine PVDF-TrFE co-polymer SEA extrusion printed film was −1.53 pm V$^{-1}$, 207% higher than that of the solvent-cast PVDF-TrFE co-polymer ($d_{33}$ = −0.50 pm V$^{-1}$). These results are consistent with reports on solvent-cast fluoropolymer films and shear stress-induced partial polarization[8,32,41,43,44]. The

$d_{33}$ of the Ti$_3$C$_2$T$_x$/PVDF-TrFE (0.50 wt%) SEA extrusion printed film increased up to a maximum at −5.11 pm V$^{-1}$, an ultimate improvement of 234% over the printed pristine PVDF-TrFE co-polymer film and 926% over the solvent-cast PVDF-TrFE co-polymer film.

Unexpectedly, considering the lower $I_\beta/I_\gamma$ ratio (Fig. 4b) and net crystallinity (Fig. 4c) of the Ti$_3$C$_2$T$_x$/PVDF-TrFE (0.02 wt%) film, an increase in the $d_{33}$ of 72% relative to the pristine PVDF-TrFE co-polymer film was observed (Fig. 5c). This would suggest that even at low Ti$_3$C$_2$T$_x$ nanosheet loadings there is a sufficient increase in induced polarization locking of the PVDF-TrFE co-polymer to offset the net decrease in the electroactive phase within the composite.

Surprisingly, the effective $d_{33}$ for the solvent-cast Ti$_3$C$_2$T$_x$/PVDF-TrFE (0.50 wt%) film matched that of the extrusion printed film with the same Ti$_3$C$_2$T$_x$ nanosheet loading (−5.11 pm V$^{-1}$), suggesting the effect of induced polarization locking from the Ti$_3$C$_2$T$_x$ nanosheet surface is favored over the shear-induced polarization with high Ti$_3$C$_2$T$_x$ nanosheet loading. It is hypothesized that this arises due to surface area minimization of the Ti$_3$C$_2$T$_x$ nanosheets during solvent casting, where the sheets settle perpendicular to gravity in order to maximize the area upon which the force is acting[27]. This result, while unintuitive, confirms that the driving force for the polarization enhancement is the fundamental interaction and local dipole locking occurring in the solution between the Ti$_3$C$_2$T$_x$ nanosheet and the PVDF-TrFE co-polymer, rather than fluoropolymer chain alignment induced through SEA 3D printing.

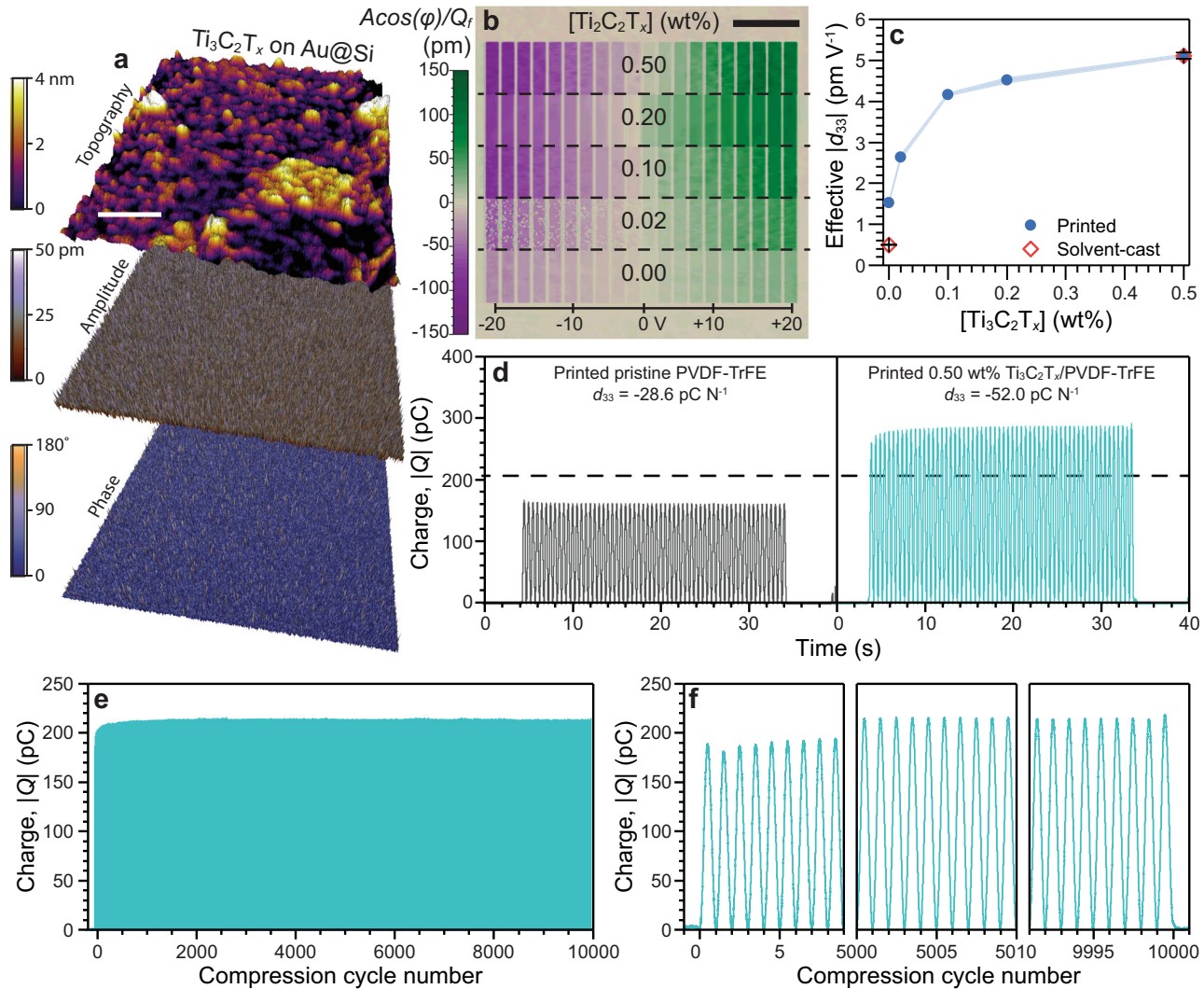

**Fig. 5 Polarization and energy harvesting of Ti₃C₂Tₓ/PVDF-TrFE composites.** a Dual AC resonance tracking (DART) PFM of a Ti₃C₂Tₓ nanosheet on gold-coated silicon (Au@Si) substrate, showing the topography (top), piezoelectric amplitude (middle), and piezoelectric phase trace (bottom). Scale bar represents 200 nm. **b** PFM of Ti₃C₂Tₓ/PVDF-TrFE (0.00 wt%, 0.02 wt%, 0.10 wt%, 0.20 wt%, and 0.50 wt%) SEA extrusion printed films, showing the piezoresponse ($Acos(\varphi)/Q_f$) under an applied voltage between −20 V and +20 V. Scale bar represents 1 μm. **c** The effective piezoelectric charge coefficient ($d_{33}$) calculated from the PFM data, including for solvent-cast PVDF-TrFE and Ti₃C₂Tₓ/PVDF-TrFE (0.50 wt%) films as controls. The error bars represent the mean ±SE. **d–f** The macroscale energy harvesting characterization of the Ti₃C₂Tₓ/PVDF-TrFE (0.00 wt% and 0.50 wt%) SEA extrusion printed PEGs with input force ($\Delta F$) at 10 N following a sinusoidal input signal. **d** The generated surface charge as a function of time for 60 compression cycles at 2 Hz. The horizontal dashed line represents the charge generated from a completely polarized ($d_{33} = -38$ pC N⁻¹) PVDF-TrFE co-polymer for a PEG with similar dimensions. **e** The stability of the generated charge as a function of cycle count over 10,000 cycles for the Ti₃C₂Tₓ/PVDF-TrFE (0.50 wt %) PEG at 10 Hz and **f** the expanded initial, middle, and end regions of the charge stability data in **e** showing 1 s of data (10 cycles) in each panel.

**Macro energy harvesting using Ti₃C₂Tₓ/PVDF-TrFE composite PEGs.** Macroscale electromechanical testing confirmed the trends in the $d_{33}$ observed by PFM and quantified the energy output from the Ti₃C₂Tₓ/PVDF-TrFE PEGs. For the measurements, the PEGs were compressed following a sinusoidal force pattern as a function of time with amplitude ($\Delta F$) at 10 N (Supplementary Fig. 26a, b), pre-loaded to 5 N to minimize artefacts from contact electrification (i.e., contact-separation and lateral-sliding triboelectric modes)[44,45]. The harvested energy was measured by a charge amplifier, negating any inherent effects of capacitance from the Ti₃C₂Tₓ/PVDF-TrFE composite films and the electrical cables, commonly unaccounted for during voltage measurements.

Macroscale $d_{33}$ measurements were performed on the SEA extrusion printed pristine PVDF-TrFE and Ti₃C₂Tₓ/PVDF-TrFE (0.5 wt%) PEGs (Fig. 5d). The $d_{33}$ of the pristine SEA-printed

pristine PVDF-TrFE PEG (−28.6 pC N⁻¹) was lower than literature reports for poled PVDF-TrFE (~−38 pC N⁻¹), however, this is not unexpected[3,30]. More importantly, the $d_{33}$ of the Ti₃C₂Tₓ/PVDF-TrFE (0.50 wt%) was significantly larger than any prior reports on unpoled fluoropolymers at −52.0 pC N⁻¹[4]. Notably, the ratio observed between the $d_{33}$ of pristine PVDF-TrFE to the Ti₃C₂Tₓ/PVDF-TrFE (0.50 wt%) via the direct piezoelectric effect (macroscale measurements, $D_3 = d_{33}\sigma_3$) of ~1:2 correlated well to the trend observed via the converse piezoelectric effect (PFM, $\gamma_3 = d_{33}E_3$) of ~1:3. The improvement in measured charge arises from the piezoelectric effect, connecting the induced local polarization locking described by MD to the macroscale polarization and subsequent energy harvesting.

The $g_{33}$ was determined from the measured dielectric permittivity ($\varepsilon_{33}$), which is shown in Supplementary Information Fig. 30. The relative permittivity was observed to increase slightly

with the addition of the $Ti_3C_2T_x$ nanosheets, from 11.6 in the pristine PVDF-TrFE film to 14.4 in the $Ti_3C_2T_x$/PVDF-TrFE (0.50 wt%) film, in close agreement with prior reports on the dielectric properties of MXene/fluoropolymer composites[46]. The $g_{33}$ of the SEA extrusion printed PEGs exhibited similar characteristics upon the addition of $Ti_3C_2T_x$ nanosheets to the PVDF-TrFE co-polymer, improving 18% at 0.50 wt% $Ti_3C_2T_x$ (402 mV m $N^{-1}$) relative to the pristine PVDF-TrFE (341 mV m $N^{-1}$). This enhancement is important, as the $g_{33}$ possesses an inverse relationship with the $\varepsilon_{33}$ and most reports enhance only one of the two piezoelectric coefficients[47]. Thus, the observed $g_{33}$ value of the 0.50 wt% $Ti_3C_2T_x$/PVDF-TrFE PEG is higher than that of literature values for electrically poled PVDF-TrFE PEGs (380 mV m $N^{-1}$)[3]. The piezoelectric figure of merit, $d_{33}g_{33}$, is consequently larger for the 0.50 wt% $Ti_3C_2T_x$/PVDF-TrFE PEG (20.9 × $10^{-12}$ $Pa^{-1}$) relative to both the pristine PVDF-TrFE PEG prepared here (9.7 × $10^{-12}$ $Pa^{-1}$) and literature values for poled PVDF-TrFE PEGs (14.4 × $10^{-12}$ $Pa^{-1}$)[3], as shown in Supplementary Information Fig. 31.

To demonstrate the stability of the PEGs, a compressive cycling stability study on the $Ti_3C_2T_x$/PVDF-TrFE (0.50 wt%) PEG was performed for 10,000 cycles with a $\Delta F = 10$ N and frequency of 10 Hz (Fig. 5e). Figure 5f shows selected regions at the beginning, middle, and end of the measurement. Aside from the initial period of cycling, which exhibited a rising generated charge from compressive stress to ~−185 pC, the measured charge remained stable for the entire cycling period at −215 pC. The stability of the generated charge infers these completely solid-state PEGs can be used for long-term energy harvesting from human motion.

## Discussion

In summary, we have developed a mechanistic understanding of how nanofillers with no out-of-plane piezoelectricity can influence the local and macroscale polarization of fluoropolymers using 2D $Ti_3C_2T_x$ nanosheets as templates. We show conclusively that $Ti_3C_2T_x$ nanosheets have a strong electrostatic interaction with the PVDF-TrFE co-polymer, resulting in the evolution of a locked polarization in the fluoropolymer, perpendicular to the basal plane of the $Ti_3C_2T_x$ nanosheet. This effect is not observed using graphene nanosheets. The unique 2D geometry of $Ti_3C_2T_x$ nanosheets means that by either SEA extrusion printing or solvent casting we can elegantly and simply translate this induced local net polarization into macroscale polarization, demonstrating an exceptional $d_{33}$ of −52.0 pC $N^{-1}$, without the need for arduous and energy-intensive electrical poling. The strong electrostatic interactions between the nanofiller and fluoropolymer resulted in a mechanically robust and flexible PEG device, capable of harvesting energy over 10,000 cycles without any degradation in performance.

Tuning surface terminations on MXenes and other 2D materials could afford enhanced electrostatic interactions, leading to further improvements in piezoelectric outputs in fluoropolymers. Leveraging this new understanding of nanoscale phenomena at the interface of a fluoropolymer and a 2D sheet now opens up a plethora of research opportunities to design piezoelectric composites with broad applicability, including in wearable energy harvesting[8], piezocatalysis[10], piezophotonics[11], and anisotropic sensors[48]. Coupled with the elegant and versatile fabrication methods, our sustainable system can enable bespoke device design in emerging fields for robotic interfaces[9], biomedical implants[9], and direct-on-organ printed electronics[37].

## Methods

**Synthesis of $Ti_3C_2T_x$ dispersion.** $Ti_3C_2T_x$ nanosheets were synthesized by selectively etching aluminum atoms out of a $Ti_3AlC_2$ (MAX phase) parent ternary carbide precursor[49]. In brief, 1 g of $Ti_3AlC_2$ powder (Carbon-Ukraine Ltd., <40

μm) was slowly added into hydrochloric acid (HCl, 20 mL, 9 M) containing lithium fluoride (LiF, 1.6 g, 99.5%). The dispersion was then stirred at room temperature for 24 h to etch out the aluminum from the $Ti_3AlC_2$ MAX phase. Afterwards, the dispersion was repeatedly centrifuged (Allegra X-12R with FX6100 fixed angle rotor, Beckman Coulter) at 1345 × g (10 min each time) and washed using ultra-pure water to raise the pH of the dispersion. When the pH approached 6, the $Ti_3C_2T_x$ suspension was probe sonicated for 10 min (2 s on and 2 s off) in an ice-bath under an argon gas flow. The dispersion was then centrifuged at 247 × g for 30 min to remove any multi-layer $Ti_3C_2T_x$ and unetched $Ti_3AlC_2$. The supernatant containing single-layer $Ti_3C_2T_x$ nanosheets was concentrated by further centrifuging at 7025 × g for 30 min and the sediment was dispersed into DMF. This process was repeated three times to prepare dispersions of $Ti_3C_2T_x$ nanosheets in DMF (at 4.4 mg $mL^{-1}$).

**MDs simulations.** All-atom MDs simulations were used to elucidate the interactions between PVDF-TrFE co-polymer films and the $Ti_3C_2T_x$ nanosheet substrate. Each PVDF-TrFE chain contained 21 VDF and 9 TrFE monomers, corresponding to molar concentrations of 70 mol% VDF and 30 mol% TrFE. The co-polymer chains were introduced at 2 nm from the substrate having an initial film density of 1.3 g $cm^{-3}$ using the materials and processes simulations 4.3 platform (Scienomics). The interactions between the $Ti_3C_2T_x$ monolayer nanosheet and the PVDF-TrFE co-polymer film were simulated in the NVT (constant number of atoms, volume, and temperature) ensemble at 298.15 K in a simulation box with periodic boundary conditions. All subsequent simulations were performed on the University of Melbourne's high-performance computing system using the large-scale atomic/molecular massively parallel simulator[50]. The equations of motion were integrated using the velocity-Verlet algorithm[51] with a time step of 1 fs using the transferable, extensible, accurate, and modular forcefield (TEAM-FF) for both the $Ti_3C_2T_x$ nanosheet and the PVDF-TrFE co-polymer. Partial charges were assigned to each atom of the PVDF-TrFE using the bond increments method, whereas the charges for the atoms of the $Ti_3C_2T_x$ surface were adopted from the first-principle calculations[52]. The interatomic potential was validated with the density ($\rho$) of bulk PVDF-TrFE melts of various molecular weights at 230 °C, converging to a plateau at ~1.42 g $cm^{-3}$ (Supplementary Fig. 2). The obtained density agreed well (deviation < 5%) with the experimental value ($\rho_{bulk}$ = 1.49 g $cm^{-3}$) of the bulk PVDF-TrFE co-polymer, validating the TEAM-FF interatomic potential used in the present MD simulations. It should be mentioned that in our simulations the $Ti_3C_2T_x$ nanosheet substrate was kept frozen, interacting with the atoms of the PVDF-TrFE co-polymer chains only via van der Waals and electrostatic interactions.

To quantify the strength of the interaction between the $Ti_3C_2T_x$ nanosheet substrate and a single PVDF-TrFE chain, individual NVT MD simulations were carried out by applying a constant force normal and opposite to the $Ti_3C_2T_x$ nanosheet surface to each atom of the PVDF-TrFE chain and gradually increasing it from 0.00 pN to 6.95 pN with a step of 0.695 pN. At each one of these simulations, we monitored the position of the co-polymer chain and recorded at which applied force the chain was fully desorbed from the $Ti_3C_2T_x$ nanosheet surface.

**Extrusion printing ink preparation.** PVDF-TrFE co-polymer powder (75 mol% VDF and 25 mol% TrFE, $M_w$ at 420 kDa, density at 1.49 g $cm^{-3}$, Solvay) and acetone (AR grade, Chem-Supply Pty Ltd) was used without further purification. The stock $Ti_3C_2T_x$ nanosheet dispersion ($Ti_3C_2T_x$ nanosheets in DMF at 4.4 mg $mL^{-1}$) was diluted in acetone to provide solutions with $Ti_3C_2T_x$ nanosheet concentrations of 0.00 mg $mL^{-1}$, 0.10 mg $mL^{-1}$, 0.52 mg $mL^{-1}$, 1.05 mg $mL^{-1}$, and 2.61 mg $mL^{-1}$. PVDF-TrFE co-polymer powder was added to the $Ti_3C_2T_x$ nanosheet/acetone solutions at 40 wt% relative to acetone and stirred at 23 °C until homogeneous inks formed containing $Ti_3C_2T_x$ nanosheet concentrations relative to PVDF-TrFE at 0.00 wt%, 0.02 wt%, 0.10 wt%, 0.20 wt%, and 0.50 wt%, respectively. The PVDF-TrFE co-polymer concentration optimization in acetone is presented in the Supplementary Information.

**Extrusion printing and PEG device fabrication.** The $Ti_3C_2T_x$/PVDF-TrFE composite inks (0.00 wt%, 0.02 wt%, 0.10 wt%, 0.20 wt%, and 0.50 wt%) were transferred into 3D printing dispensing barrels (30 mL, Optimum, Nordson EFD), then sealed and stored at −5 °C prior to printing. A 3D printer (Bioplotter 3D, Envisiontec) was preset to print a single-layer square film (3 cm × 3 cm) onto a clean glass plate substrate using a raster pattern with a line spacing of 350 μm. During printing, the cartridge containing the ink was kept at 5 °C and the glass plate substrate at 23 °C, the xy speed was 30 mm $s^{-1}$ and extrusion pressure was 1.7 bar through a tapered nozzle with an internal diameter of 200 μm (SmoothFlow, Nordson EFD). After printing the films were placed in a vacuum oven set at 23 °C for 20 min to ensure the complete extraction of the solvent.

To fabricate the PEGs, the SEA extrusion printed films were removed from the glass plate and coated with a seeding layer of chromium (Cr) and an electrode layer of gold (Au) (total thickness of 60 nm) via sputter deposition (Nanochrome I, Intlvac Thin Film Corporation) through a shadow mask on both sides of the film with a total overlapping electrode area of 2.4 $cm^2$. Wires (FLEXI-E 0.15, Stäubli Electrical Connectors AG) were soldered to copper (Cu) foil with conductive

adhesive (1181, 3 M) and adhered to each side of the printed film without overlap, ensuring strong contact between the Cu tape and Cr/Au coating. The printed films were finally encapsulated in insulating Kapton polyimide tape on both surfaces to complete the PEGs. The extended fabrication details are presented in the Supplementary Information.

## Materials characterization

*Rheology.* The rheological characterization of the pristine PVDF-TrFE co-polymer and the $Ti_3C_2T_x$/PVDF-TrFE (0.20 wt%) inks in acetone was undertaken using a strain-controlled rheometer (MCR 702, Anton Paar) with cone-plate geometry at 5 °C. The gap was set at 102 µm, with a cone diameter of 25 mm and an angle of 2°. Frequency and shear strain measurements were undertaken in oscillatory mode with an upwards logarithmic ramp of the frequency under fixed shear strain at 1% and shear strain under fixed frequency at 1 Hz, respectively. The printing simulation measurement was performed in oscillatory mode with the frequency fixed at 1 Hz and controlled shear stress. Initially, 1 Pa shear stress was applied to the inks for 50 s to demonstrate the properties of the inks at rest. Subsequently, 6 kPa shear stress was applied to the sample for 70 s, followed by 1 Pa for 70 s to demonstrate recovery of the ink, repeated for an additional high shear stress cycle.

*Helium ion microscopy.* The surface of the SEA extrusion printed $Ti_3C_2T_x$/PVDF-TrFE (0.50 wt%) film was imaged using helium ion microscopy (Orion NanoFab, Zeiss) in order to determine the orientation and distribution of the $Ti_3C_2T_x$ nanosheets in the PVDF-TrFE co-polymer. The micrographs were obtained with a 100 µm field of view using a dwell time of 0.5 µs and an accelerating voltage of 30 kV.

*Tensile testing.* Tensile tests were performed on a mechanical tester (Electroforce 5500, Bose). The $Ti_3C_2T_x$/PVDF-TrFE SEA extrusion printed films (0.00 wt%, 0.02 wt%, 0.10 wt%, 0.20 wt%, and 0.50 wt%) were cut into strips (27 mm × 5 mm) and mounted in the grips, with an exposed length at 5 mm. The samples, with average thickness at 45 µm, were extended at 0.01 mm s$^{-1}$ up to a maximum of 11.4 mm.

*Crystallinity.* The crystallinity of the $Ti_3C_2T_x$/PVDF-TrFE SEA extrusion printed films (0.00 wt%, 0.02 wt%, 0.10 wt%, 0.20 wt%, and 0.50 wt%) was obtained using DSC. The samples were placed in a ceramic crucible at 25 °C and heated at 10 °C min$^{-1}$ to 200 °C under a nitrogen gas flow (20 mL min$^{-1}$). The crystallinity calculation methods are presented in the Supplementary Information.

*Vibrational spectroscopy.* The distribution of the $Ti_3C_2T_x$ nanosheets in the PVDF-TrFE co-polymer, along with the β:γ intensity ratio was confirmed using Raman confocal microscopy (inVia, Renishaw), equipped with a 532 nm laser, an 1800 line mm$^{-1}$ grating, and a ×50 objective. Typical maps were obtained for a surface area of 20 µm length and width, at a pixel density of 1 px µm$^{-1}$. Each pixel of the map consisted of a spectrum, centered at 1300 cm$^{-1}$ and obtained with an exposure time of 0.15 s accumulated over 1000 scans.

*Piezoresponse force microscopy.* Nanoscale polarization of the $Ti_3C_2T_x$/PVDF-TrFE SEA extrusion printed films (0.00 wt%, 0.02 wt%, 0.10 wt%, 0.20 wt%, and 0.50 wt%) was measured using an Asylum Research Cypher ES atomic force microscope. The data were obtained at 23 °C in the air using conductive Pt cantilevers (12PT400B, Rocky Mountain Nanotechnology) with spring constant at 0.3 N m$^{-1}$ and tip radius below 20 nm. The scans were undertaken on an area of 5 µm × 5 µm at 256 pixels per line, corresponding to ~20 nm per pixel. The PFM was carried out in contact lithography, whereby a potential was applied by the tip to the sample following a pre-defined pattern, observing the amplitude and phase changes relative to the unbiased areas. The applied potential ranged between −20 V and +20 V in increments of 2 V on each scan line at a scan rate of 0.2 Hz, with each voltage step applied to a total area of 0.195 µm in the x direction and 3.906 µm in the y direction. The resultant amplitude and phase values were then correlated to the corresponding voltage for each pixel of the scan. Extended details of the PFM technique are presented in the Supplementary Information.

*Macroscale electromechanical quantification.* Macroscale polarization of the $Ti_3C_2T_x$/PVDF-TrFE PEGs (0.00 wt% and 0.50 wt%) was measured via the input of cyclic compressive force and measurement of the resulting surface charges. The force was applied by a mechanical tester (Electroforce 5500, Bose) to the active area of the PEG using a sinusoidal waveform with frequency at 2 Hz, the minimum force set at 5 N, and maximum force set at 15 N, corresponding to $\Delta F$ at 10 N and stress at 25 kPa. The surface charge was measured with a charge amplifier (Nexus 2692, Brüel & Kjær) and recorded through a data acquisition system (9223, National Instruments). The cycling stability testing was undertaken at 10 Hz. The extended data and discussion are presented in the Supplementary Information.

## Data availability

The data that support the findings of this study are available from the corresponding author upon reasonable request.

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

## Acknowledgements

This research was supported by the Australian Government through the Australian Research Council's Linkage Projects funding scheme (LP160100071), Future Fellowships funding scheme (FT130100380), and Industry Transformation Research Hub funding scheme (IH140100018). This work was performed in part at the Materials Characterization and Fabrication Platform (MCFP) at the University of Melbourne and the Melbourne Centre for Nanofabrication (MCN) in the Victorian Node of the Australian National Fabrication Facility (ANFF). The authors wish to thank Dr. James Bullock and Dr. Brett Johnson for performing the dielectric characterization, and Robert Delaney for helpful discussions on energy harvesting methods.

## Author contributions

Conceptualization, N.A.S., P.C.S., J.M.R., and A.V.E.; methodology, N.A.S., E.N.S., E.G., and J.Z.; formal analysis, N.A.S., P.C.S., E.N.S., and E.G.; investigation, N.A.S., P.C.S., E.N.S., E.G., J.Z., V.C.L., B.I., and K.A.S.U.; data curation, N.A.S.; writing—original draft, N.A.S., P.C.S., and A.V.E; writing—review and editing, all authors; visualization, N.A.S., P.C.S., E.N.S., P.C.S., J.Z., and B.I.; supervision, A.V.E., J.M.R., G.W.D., and J.G.S.; project administration, A.V.E., G.W.D., J.G.S., and J.M.R.; funding acquisition, A.V.E., G.W.D., J.G.S., and J.M.R.

## Competing interests

The authors declare no competing interests.
