## [Peer Review File · Nature Communications]

REVIEWER COMMENTS

Reviewer #1 (Remarks to the Author):

Very nice paper on use of mxene based filler for use with PVDF - there is evidence that electrostatic interaction of PVDF with the filler leads to poling. Alignment with extrusion or other methods leads to a polarisation of the material for sensing or energy harvesting. The characterisation and discussion are in depth and clearly articulated. I have no negative comments. The supplemental details are also detailed. it could benefit from some macro-scale polarisation field loops to see if some switching can be achieved or not. For the d33 it could also be beneficial to measure with conventional d33 meter as opposed to the method in the paper. Nevertheless, excellent work.

Reviewer #2 (Remarks to the Author):

The current method of inducing net polarization in the piezoelectric fluoropolymers is an electrical poling process which is energy-intensive. In order to eliminate this process and enable the low-energy production of efficient energy harvesters, the authors revealed a hitherto unseen polarization locking phenomena of PVDF-TrFE perpendicular to the basal plane of two-dimensional (2D) Ti3C2Tx MXene nanosheets. The measured piezoelectric charge coefficient, d33, of -52.0 pC/N is significantly higher than the current PVDF-TrFE (-38 pC/N). The study of this manuscript provides a new fundamental and low energy input mechanism of poling fluoropolymers. To further improve the quality of this manuscript, the authors are suggested to address concerns as listed in the following:

1. The measured piezoelectric charge coefficient d33 of -52 pC/N is very impressive. But for piezoelectric materials, d31 is also an important property. The d31 of current PVDF-TrFE ranges from 25 to 35 pC/N. If the measured d31 is also much higher than the other PVDF materials, the results of this manuscript can be more solid. Because for some kind of applications, such as pMUTs and cantilever actuators, d31 is more crucial than d33.
2. Besides d31 and d33, other electric properties, such as piezo stress constant (g_{31} & g_{33} , Vm/N), dielectric constant (ϵ_r), pyroelectric coefficient (p , C/m²K), electromechanical coupling factor (k_{31} & k_{33} , %), maximum voltage (E , V/ μ m) and so on should be measured. Besides Young's modulus, other mechanical properties such as tensile strength (σ_{MD} & σ_{TD} , N/m²) and so on should be measured.
3. The authors take the energy harvester as an example to reveal the performance Ti3C2Tx/PVDF-TrFE films. How about fabricating an energy harvester with traditional PVDF-TrFE film and comparing their performance? And the authors would better fabricate some PVDF cantilever actuators and test their bending magnitude under the same voltage. Those works will significantly enrich the manuscript.
4. In Page 12 Line 5 and Fig. 2 i, the Young's modulus of the PVDF-TrFE co-polymer film is smaller than the literature values. Because Young's modulus will impact the performance of PVDF based sensors and actuators, more clarifications are needed. In Fig. 2 g, h, and i, it seems that the wt% of Ti3C2Tx has a relationship with those mechanical properties (Tensile strain (γ_t)-stress (σ_t) profiles and Young's modulus), more details about this phenomenon are needed.
5. In order to improve the deformation and output force, some commercial PVDF films are stretched at the last fabrication step. So if the proposed Ti3C2Tx/PVDF-TrFE films are stretched, what will happen? For different wt% of Ti3C2Tx, what's the difference?
6. For piezoelectricity, one important factor that should be discussed is the piezo-domain without electrical poling. The authors have largely discussed the influence of electrical poling but don't mention the effect of piezoelectricity. A brief discussion between domains and dipole moments should be included correspondingly to the bulk co-polymer and Nanosheets.

7. For Fig 1a, the authors have demonstrated a comparison between the graphene substrate and Ti₃C₂T_x substrate. A brief explanation in the particular reason for graphene as a reference should be added. Then, the authors discussed the microscopic force between the first layer and the second layer. However, more explanations should be carried out according to the surface energy and bond interactions. As more chains fill the free space on the Ti₃C₂T_x substrate, the microstructure will change between the first and second layers. Correspondingly, surface relaxation and reconstruction etc. may also happen. A more detailed discussion about morphology and interactions should be included in order to reach the result of electrostatic force, bond conformations, and electroactive phase.

8. For the mechanical property part of the SEA extrusion printed Ti₃C₂T_x/PVDF-TrFE, the authors have demonstrated the ductility from yield point and material elongations. A homogenous dispersion result is reached due to the linear increase in Young's modulus. However, since the slope of the stress-strain curve is proportional to elastic modulus and elastic modulus is also proportional to the slope of force curve dF/dr at equilibrium distance, a more detailed explanation between homogenous dispersion and bond strength should be included based on the result of elastic modulus. Moreover, as a material is elastically elongated under a tensile load, there is a change in Poisson's ratio. An all-rounded discussion corresponding to lateral and longitudinal is suggested.

9. The authors have discussed the different PVDF-TrFE phases which is determinant in crystallinity and energy harvesting capabilities. If possible, a microscopic morphology schematic of each phase and compositions is helpful to demonstrate the result and relationships.

10. The whole manuscript is based on fluoropolymer; however, the authors haven't mentioned the effect of temperature (e.g., temperature-dependent relaxations on the surface between Nanosheet and bulk copolymer, polymer temperature-dependent phases, temperature influence on trans or gauche compositions, etc.). Discussing the influence of temperature would be valuable for the manuscript.

Reviewer #3 (Remarks to the Author):

I reviewed the paper by Shepelin et al. on the piezoelectric polarisation formed between PVDF-TrFE and Ti₃C₂T_x. The paper is interesting, shows some timely results involving simulations and measurements, and I believe that it may have an impact in the field. I'd have few questions as shown below:

1) According to the data, one of the main driving forces for the stabilisation of the polarisation vector perpendicular to the surface is the strong electrostatic interactions between the macro-molecule and Ti₃C₂T_x. However, I have not seen any simple charge density plots that would show clearly this effect. I also checked the SI but nothing there. I'd suggest the author to show a few charge density difference plots which will help to set down the idea of the strong electrostatic interactions. At the moment it looks indirect though. We can see the effect in terms of the polarisation vector but we don't see the main players creating it.

2) On page 7, it is mentioned that the electrostatic screening length in 2D materials is 1-10 nm, no citations or references are included whatsoever. Moreover, there is some speculation on the origin of the polarisation locking via electrostatic forces, which is related with comment 1) above, but not solid data is shown. I'd suggest the author to undertake a careful analysis on the clarification of the origin of the locking effect before stating anything else.

3) It has been show recently by Tian et al. Nano Letters 2020, 20, 2, 841–851 that the electronic polarisability is the fundamental variable for the dielectric properties of 2D materials. If electrostatic interactions are related to the locking phenomena, then I'd assume that the strong anisotropy

between both kind of interactions (in-plane, out-of-plane) will play a role in the effect. This is also related with the short interactions $\sim 5 \text{ \AA}$ perpendicular to the layer mentioned by the authors (page 7), but not fully referenced. I would suggest the authors to calculate the range of interactions involved in the interactions between Ti3C2Tx and the macro-molecule using the methods in Tian et al. This will give a good idea about the length where the interaction dying out in the system.

4) On page 8, it is also mentioned the local polarisation but no data included to unveil its origin. Please improve.

5) Can the authors show some histogram of the amount of devices fabricated and their performance? This is very important to proof the accuracy of the device-data, and not just one device worked out fine.

6) Similar to 5), how many samples were used to extract the value of d_{33} ? Some histogram would be helpful to have some statistics of its magnitude.

7) The ink part is very interesting, but I missed some discussions whether other solvents were tried or utilised in the process of fabrication the devices.

8) Regarding degradation, are the films stable under radiation e.g. light, laser power, etc. Some comments/data regarding external driving forces used to characterise the samples would be helpful for further verification from other groups.

DETAILED RESPONSE TO REVIEWERS

The authors would like to thank the editors for their time, expertise, and dedication in reviewing our manuscript, please find below a detailed point-by-point response to the reviewers' comments and recommendations.

Reviewer #1 (Remarks to the Author):

Very nice paper on use of mxene based filler for use with PVDF - there is evidence that electrostatic interaction of PVDF with the filler leads to poling. Alignment with extrusion or other methods leads to a polarisation of the material for sensing or energy harvesting. The characterisation and discussion are in depth and clearly articulated. I have no negative comments. The supplemental details are also detailed. it could benefit from some macro-scale polarisation field loops to see if some switching can be achieved or not. For the d_{33} it could also be beneficial to measure with conventional d_{33} meter as opposed to the method in the paper. Nevertheless, excellent work.

We would like to thank the reviewer for their positive comments. It is heartening to read such an appreciative review.

We agree on the general utility of the polarization field loops in the characterization of the ferroelectric properties. However, the application of bias (E) above the coercive field ($E > E_c$) would influence the polarization status of the material, which is detrimental for non-destructive measurement of the polarization status.² We have avoided this limitation by utilizing PFM below the E_c of PVDF-TrFE ($< 50 \text{ MV m}^{-1}$)³ for the comparative measurement of the converse piezoelectric effect.

In regards to using a conventional d_{33} meter, it is worth noting that our macroscale d_{33} measurements were obtained using the Berlincourt (quasi-static) method,¹ which directly replicates a d_{33} meter in its operational principle. Through this measurement technique, we have directly measured the electric displacement (D_3) as the fundamental output value arising from the cyclic application of stress (σ_3). Here, $D_3 = Q/A_D$, where Q is the generated surface charge and A_e is the overlapping electrode layer, and $\sigma_3 = F/A_\sigma$, where F is the applied force and A_σ is the impacted area. Thus, our measurement in the manuscript is perfectly suitable without the need for comparison to a conventional meter.

Reviewer #2 (Remarks to the Author):

The current method of inducing net polarization in the piezoelectric fluoropolymers is an electrical poling process which is energy-intensive. In order to eliminate this process and enable the low-energy production of efficient energy harvesters, the authors revealed a hitherto unseen polarization locking phenomena of PVDF-TrFE perpendicular to the basal plane of two-dimensional (2D) Ti₃C₂T_x MXene nanosheets. The measured piezoelectric charge

coefficient, d_{33} , of -52.0 pC/N is significantly higher than the current PVDF-TrFE (-38 pC/N). The study of this manuscript provides a new fundamental and low energy input mechanism of poling fluoropolymers.

We thank the reviewer for their detailed reading and understanding of the manuscript.

To further improve the quality of this manuscript, the authors are suggested to address concerns as listed in the following:

1. The measured piezoelectric charge coefficient d_{33} of -52 pC/N is very impressive. But for piezoelectric materials, d_{31} is also an important property. The d_{31} of current PVDF-TrFE ranges from 25 to 35 pC/N. If the measured d_{31} is also much higher than the other PVDF materials, the results of this manuscript can be more solid. Because for some kind of applications, such as pMUTs and cantilever actuators, d_{31} is more crucial than d_{33} .

We thank the reviewer for their helpful comments. We agree with the reviewer that for certain applications the d_{31} coefficient is important. However, the d_{33} provides the highest piezoelectric coefficient in PVDF-TrFE, hence we paid special attention to this coefficient in order to maximize it. This has also been observed in literature.

Furthermore, from a practical perspective, the d_{31} and d_{33} are interconnected during measurements *via* the Poisson's ratio. Notably, the thickness of our films in the 3 direction (10^{-5} m) was significantly lower than that of the 1 direction (10^{-2} m), therefore the effect of thickness-axis compression will induce a lower elongation in the 1 direction compared to the induced thickness-axis compression from elongation in the 1 direction. Here, we believe the measurement of d_{31} would incorporate significant influences from the d_{33} due to the Poisson's ratio influence.

2. Besides d_{31} and d_{33} , other electric properties, such as piezo stress constant (g_{31} & g_{33} , Vm/N), dielectric constant (ϵ_r), pyroelectric coefficient (ρ , C/m²K), electromechanical coupling factor (k_{31} & k_{33} , %), maximum voltage (E , V/ μ m) and so on should be measured. Besides Young's modulus, other mechanical properties such as tensile strength (σ_{MD} & σ_{TD} , N/m²) and so on should be measured.

We thank the reviewer for their comment and agree that these values are important to measure for a new composite material system. To this end we have added in calculations of the dielectric constant (ϵ_r), piezoelectric voltage coefficient (g_{33}) and the piezoelectric figure of merit (FOM). The expanded discussion of the information presented below has been inserted into the Supplementary Information.

We have measured the average ϵ_r of the SEA extrusion printed films, finding a slight increase from 11.6 in the pristine PVDF-TrFE film to 14.4 in the 0.50 wt% $Ti_3C_2T_x$ /PVDF-TrFE film. The average ϵ_r as a function of frequency for the SEA extrusion printed pristine PVDF-TrFE and 0.50 wt% $Ti_3C_2T_x$ /PVDF-TrFE films, as well as the ϵ_r at 100 Hz as a function of $Ti_3C_2T_x$

concentration in PVDF-TrFE, have been inserted into the Supplementary Information and are shown in Figure R1 below for convenience.

Fig. R1: Dielectric properties of the SEA extrusion printed films. **a** The frequency dependence of the dielectric constant (ϵ_r) for the pristine PVDF-TrFE copolymer film and the 0.50 wt% $\text{Ti}_3\text{C}_2\text{T}_x/\text{PVDF-TrFE}$ film. **b** The $\text{Ti}_3\text{C}_2\text{T}_x$ concentration dependence of the ϵ_r obtained at 100 Hz. The error bars correspond to data collected for three separate films.

Following the determination of ϵ_r , we have then calculated the macroscale g_{33} via $g_{33} = d_{33}/\epsilon_{33}$ ($\epsilon_{33} = \epsilon_r \epsilon_0$, $\epsilon_0 \approx 8.854 \times 10^{-12} \text{ F m}^{-1}$)² for the pristine PVDF-TrFE and the 0.50 wt% $\text{Ti}_3\text{C}_2\text{T}_x/\text{PVDF-TrFE}$ films, observing an 18% increase upon the incorporation of the $\text{Ti}_3\text{C}_2\text{T}_x$ nanosheets into the PVDF-TrFE co-polymer. Notably, following the measurement of both the d_{33} and g_{33} coefficients, we are able to calculate the figure of merit (FOM, $d_{33}g_{33}$), which has recently been demonstrated as a more universal property in comparison to the k_{33} .⁶ The FOM corresponding to the data presented in Figure 5d (Main Text) has been inserted in the Supplementary Information and below in Figure R2 for convenience.

Fig. R2: The piezoelectric figure of merit (FOM) for the SEA extrusion printed pristine $\text{Ti}_3\text{C}_2\text{T}_x/\text{PVDF-TrFE}$ PEGs, demonstrated for the pristine PVDF-TrFE (0.00 wt% $\text{Ti}_3\text{C}_2\text{T}_x$, gray circles) and the 0.50 wt% $\text{Ti}_3\text{C}_2\text{T}_x$ (teal circles) PEGs.

Each data point corresponds to one compression cycle for the data presented in Figure 5d. The horizontal bars represent the average for 60 compression cycles. The dashed blue line corresponds to the literature value for the FOM of electrically poled PVDF-TrFE.

We have also compared the piezoelectric coefficients obtained in this study with similar recent publications reporting piezoelectric coefficients. The table, located in the Supplementary Information, and in Table R1 below for convenience, shows the comparison of the FOM between our materials and those found in literature.

Table R1: Comparison of the piezoelectric charge coefficient (d_{33}), piezoelectric voltage coefficient (g_{33}) and the piezoelectric figure of merit (FOM) between the SEA extrusion printed films presented in the manuscript and fluoropolymer-based materials found in literature for which the d_{33} and g_{33} was presented.

Material	Processing	Poling	$ d_{33} $ (pC N ⁻¹)	g_{33}^{\dagger} (mV m N ⁻¹)	FOM $_{\ddagger}$ (x 10 ⁻¹² Pa ⁻¹)	Ref.
PVDF-TrFE	Extrusion printing	None	29	341	9.7	This work
0.50 wt% Ti₃C₂T_x/PVDF-TrFE	Extrusion printing	None	52	402	20.9	This work
ZnO/PVDF*	Drop casting	None	50	219	11.0	7
MnO ₂ /PVDF	Electrospinning, hot pressing and rolling	80 MV m ⁻¹ , 80 °C, 2 h	38	318	12.1	8
BCZT50/PVDF*	Melt mixing	20 MV m ⁻¹ , 70 °C, 0.5 h	27	114	3.1	9
PVDF-TrFE	Spin coating	200 MV m ⁻¹	22	187	4.0	10
PVDF	Solvent casting + double rolling	170 MV m ⁻¹ , 70 °C, 1.2 h	29	234	6.8	11
MWCNT/PVDF	Solvent casting + double rolling	100 MV m ⁻¹ , 70 °C, 1.5 h	33	169	5.6	12
PVDF-HFP	Blade coating	0.5 MV m ⁻¹ , 100 °C, 0.5 h	11	113	1.2	13

PVDF-TrFE	Solvent casting 105 MV m ⁻¹ , 23 100 °C	236	5.4	14
BT/PVDF-TrFE	Solvent casting 10 MV m ⁻¹ , 34 110 °C, 0.5 h	37	1.3	15

† Calculated *via* $g_{33} = d_{33}/\epsilon_{33}$, whereby ϵ_{33} was determined at 100 Hz;

‡ Calculated *via* $FOM = d_{33}g_{33} = d_{33}^2/\epsilon_{33}$;

* Additive has been reported to exhibit piezoelectric properties;

In regards to the measurement of tensile strength, the instrument (TA Instruments ElectroForce 5500) used to undertake the measurement of Young's modulus was limited in its range of extension (approximately 11 mm).¹⁶ During our tensile testing, we chose 5 mm as the starting separation between the tensile grips, with relative error at approximately 5% (approximately 0.25 mm error). Reducing the initial separation between the grips to 0.8 mm (whereby the maximum strain reaches 1375% following the empirical strain at break in Supplementary Information Figure S13c), an absolute error in the separation of the grips of 0.25 mm would increase the relative error to 31.3%. Hence, we decided against the reduction of the separation between grips in order to ensure reliable measurements.

The measurement of pyroelectric properties, while generally useful, is outside the scope of the electromechanical energy conversion investigated in this manuscript.

3. The authors take the energy harvester as an example to reveal the performance Ti₃C₂T_x/PVDF-TrFE films. How about fabricating an energy harvester with traditional PVDF-TrFE film and comparing their performance? And the authors would better fabricate some PVDF cantilever actuators and test their bending magnitude under the same voltage. Those works will significantly enrich the manuscript.

The authors thank the reviewer for their comments. Figure 4d provides a comparison of PVDF-TrFE energy harvesting performance produced in the same 3D printing method. In Figure 4d the pristine PVDF-TrFE co-polymer film exhibits a lower energy conversion efficiency relative to literature values of electrically poled PVDF-TrFE and the Ti₃C₂T_x/PVDF-TrFE film, as was expected due to the lack of polarization in the absence of Ti₃C₂T_x. Furthermore, we have previously described a thorough overview of the minor dipole alignment produced through extrusion printing PVDF-TrFE in a previous manuscript, also referenced in this current manuscript.¹⁷

We agree that cantilever actuators are a key application in piezoelectric devices. However, single-layer compliant materials (i.e., polymers such as PVDF-TrFE), similar to those presented in this manuscript, are not well-suited for use in cantilever systems unless they are directly bonded to additional layers, which act as high-Young's modulus substrates in order to increase the rigidity of the cantilever structure.¹⁸ Our SEA extrusion printed Ti₃C₂T_x/PVDF-TrFE films are extremely flexible (as shown in Figure R3 below) and thus are not well-suited

to cantilever systems. The direct measurement of mechanical-to-electrical energy conversion, as presented in this manuscript, provides a significantly broader scope than a single cantilever-based device morphology in the context of mechanical energy harvesting.

Fig. R3: Flexibility of the SEA extrusion printed 0.50 wt% $\text{Ti}_3\text{C}_2\text{T}_x/\text{PVDF-TrFE}$ film. **a** Side-on view of the initial state of the film. **b** The film rolled up. **c-d** The film relaxes to its initial state after being rolled up, with **c** the side-on view and **d** the top view.

4. In Page 12 Line 5 and Fig. 2 i, the Young's modulus of the PVDF-TrFE co-polymer film is smaller than the literature values. Because Young's modulus will impact the performance of PVDF based sensors and actuators, more clarifications are needed. In Fig. 2 g, h, and i, it seems that the wt% of $\text{Ti}_3\text{C}_2\text{T}_x$ has a relationship with those mechanical properties (Tensile strain (γ)-stress (σ) profiles and Young's modulus), more details about this phenomenon are needed.

The reviewer is correct in noting that the mechanical properties of our printed PVDF-TrFE film are lower than PVDF-TrFE films produced using other methodologies. We attribute this difference to the following two points:

1. The extrusion printing method uses a raster pattern, and as we are only printing a single layer there is a slight (<10%) variation in thickness between adjacent extruded sections, as shown in Figure R4 below. When printed from acetone with the substrate at room temperature, the $\text{Ti}_3\text{C}_2\text{T}_x/\text{PVDF-TrFE}$ films were observed to fully dry within 25 s from the beginning of deposition. Notably, each individual filament was found to be dry within 5 s following extrusion from the nozzle, and the entire 3 cm x 3 cm film was deposited in 20 s, hence the final strand of the extruded filament was dry 25 s after the beginning of extrusion.

Fig. R4: Simplified schematic showing the single-layer film SEA extrusion printing process for films. The film is printed following a raster pattern, whereby the nozzle orifice geometry (circular) produces circular filaments on the glass substrate (top, shown end on). The optimized rheology results in the filaments fusing together over time as the solvent evaporates (middle). The subsequent dry film (bottom) contains a periodic roughness from the filament-based printing technique.

These thickness variations of approximately $2\ \mu\text{m}$ leads to local stress concentrators during tensile testing and can lead to fracture of the film at values significantly below the fundamental strength of the PVDF-TrFE molecule. Films produced through solvent casting or homogenous production techniques would not be expected to exhibit these decreased mechanical properties.

2. Mechanical properties of films reported in literature are often of systems which have undergone some sort of cold-drawing process which significantly toughens the fluoropolymer films.

The reviewer has correctly identified that there is a relationship with increasing $\text{Ti}_3\text{C}_2\text{T}_x$ concentration and improved mechanical properties. This is anticipated due to the high Young's Modulus, and Toughness of the $\text{Ti}_3\text{C}_2\text{T}_x$ sheets coupled to the exceptional electrostatic binding between the $\text{Ti}_3\text{C}_2\text{T}_x$ nanosheets and the PVDF-TrFE co-polymer. This strong binding, as evidenced by MD calculations (Force = 4.17 pN to desorb a single PVDF-TrFE chain, page 5, line 18 of the Main Text), means that there is effective load transfer between the components. Thus, the strength of the $\text{Ti}_3\text{C}_2\text{T}_x$ nanosheets is effectively transferred to the bulk film even at exceptionally low concentrations.

5. In order to improve the deformation and output force, some commercial PVDF films are stretched at the last fabrication step. So if the proposed $\text{Ti}_3\text{C}_2\text{T}_x$ /PVDF-TrFE films are stretched, what will happen? For different wt% of $\text{Ti}_3\text{C}_2\text{T}_x$, what's the difference?

The reviewer has rightfully raised the point of the mechanical drawing treatment in the manufacture of electroactive fluoropolymers, which is commonplace in the production of PVDF homopolymer films. The motivation for this step is the increase of the f_3 (all-*trans*, highest dipole moment) phase fraction, as the f_3 phase fraction of the as-deposited PVDF homopolymer can be as low as 10%. In fact, the Piezo Film Sensors Technical Manual by Measurement Specialties, Inc.¹⁹ (one of the few commercial producers of piezoelectric

fluoropolymer films) explicitly states that co-polymer films of PVDF (e.g., PVDF-TrFE) should not undergo mechanical drawing (emphasis ours):

“Rolls of piezo film are produced in a clean room environment. The process begins with the melt extrusion of PVDF resin pellets into sheet form, followed by a stretching step that reduces the sheet to about one-fifth its extruded thickness. Stretching at temperatures well below the melting point of the polymer causes chain packing of the molecules into parallel crystal planes, called “beta phase”. To obtain high levels of piezoelectric activity, the beta phase polymer is then exposed to very high electric fields to align the crystallites relative to the poling field. Copolymers of PVDF are polarizable without stretching.”

In this manuscript, the electroactive ($\beta+\gamma$) phase fraction of the PVDF-TrFE was unchanged in the bulk with the incorporation of $\text{Ti}_3\text{C}_2\text{T}_x$ nanosheets, as observed by FTIR spectroscopy (Supplementary Information Figure S16b), shown in Figure R5b below for convenience.

Fig. R5: ATR-FTIR spectroscopy characterization of SEA extrusion printed $\text{Ti}_3\text{C}_2\text{T}_x$ /PVDF-TrFE (0.00 wt%, 0.02 wt%, 0.10 wt%, 0.20 wt% and 0.50 wt%) films. **a** The FTIR spectra, offset for clarity. **b** The relative fractions of the electroactive ($\beta+\gamma$) phases and the α phase as a function of $\text{Ti}_3\text{C}_2\text{T}_x$ concentration.

The electroactive phase fraction in our tests was found to be above 80%, thereby we expect minimal utility from mechanical drawing to offset the energy input of the process. Importantly, one of the main points of this manuscript is the orientation of the existing dipole moments of PVDF-TrFE by the addition of $\text{Ti}_3\text{C}_2\text{T}_x$ without altering either the crystallinity or the phase fraction distribution.

6. For piezoelectricity, one important factor that should be discussed is the piezo-domain without electrical poling. The authors have largely discussed the influence of electrical poling but don't mention the effect of piezoelectricity. A brief discussion between domains and dipole moments should be included correspondingly to the bulk co-polymer and Nanosheets.

We thank the reviewer for their comment. We believe we have extensively discussed the alignment of the dipole moments (or piezo-domains) without electrical poling. Namely, the work presented in this manuscript does not claim to create new dipole moments (conversion of the α phase to the β and γ phases), but rather utilizes a nanoscale template—which aligns the

existing (but randomly oriented) dipole moments of the β and γ phase parallel to the 3 directionality through electrostatic interactions at the interface—in order to maximize the piezoelectric energy conversion capabilities without electrical poling. In order to visualize the process we are demonstrating in this manuscript, we have included a simplified schematic of the polarization locking process relative to the conventional electrical poling process, both in the Main Text and in Figure R6 below for convenience.

Fig. R6: Simplified schematic outlining the nanomaterial-induced polarization locking mechanism in PVDF-TrFE as an alternative to the conventional electrical poling method. **a** The β phase PVDF-TrFE chains, obtained directly following film deposition, exhibit a randomized dipolar orientation (green arrows), resulting in a negligible net polarization (P). **b** In the electrical poling method, electrodes are attached to the surfaces perpendicular to the desired polarization direction and an electric field (E) is applied, significantly higher than the coercive field (E_c) in order to orient the individual dipole moment vectors and maximize the P to the spontaneous polarization (P_s). **c** Following the removal of the E , the PVDF-TrFE chains undergo partial relaxation from P_s to the remnant polarization (P_r). **d** Conversely, adding the $Ti_3C_2T_x$ ($T_x \approx OH$) nanosheets to the PVDF-TrFE in solution enables the P to align, without an applied electric field, perpendicular to the basal plane of the $Ti_3C_2T_x$ nanosheets via electrostatic interactions at the interface. **e** Following deposition, the $Ti_3C_2T_x$ nanosheets are generally aligned parallel to the substrate, and the subsequent evaporation of the solvent locks the P at P_s with no relaxation.

Here, the as-deposited PVDF-TrFE (panel a, with direction 1 into the page) possesses randomized individual dipole moment vector directionalities (μ) within the electroactive phases. When the individual dipole moment vector directionalities are taken into account through the Supplementary Information Equation S14, $P = \sum \mu / V$, the dipolar polarization, P , is approximately zero. The traditional poling process, shown schematically in panels b and c, applies a large electric field (>50 MV m^{-1})^{3,8,10} between the two conformally adhered electrodes at elevated temperatures (>60 °C)^{8,9} in an insulating fluid (to avoid dielectric breakdown).

Importantly, when the electric field is turned off, the partial relaxation of the dipole moment vectors results in a polarization slightly below the maximum polarization.

This manuscript demonstrates an alternative mechanism of reorienting existing dipole moment vectors within the electroactive phases of PVDF-TrFE through electrostatic interactions at the interface between the PVDF-TrFE and carefully selected nanomaterials (panels d and e). This method eliminates the requirement of an elevated temperature, an applied electric field and the attachment of electrodes for the poling step, as well as the time taken to pole the material. We have previously observed a similar effect with single-walled carbon nanotubes, however we were unable to pinpoint the origin of the polarization enhancement effect.²⁰ In the polarization locking mechanism between $Ti_3C_2T_x$ nanosheets (primary surface termination, T_x , is OH) and PVDF-TrFE copolymer in solvent (acetone), the electrostatic interactions at the interface (approximately 2 nm from the $Ti_3C_2T_x$ surface and above) reorient the PVDF-TrFE copolymer dipole moment vectors perpendicular to the basal plane of the $Ti_3C_2T_x$ (panel d). As this locking occurs in solution, the polarization remains locked to the $Ti_3C_2T_x$ when the solvent evaporates (panel e). Notably, the fast evaporation kinetics of acetone, coupled with the general tendency of the 2D nanosheets to orient their basal plane perpendicular to the gravitational force,²¹ enable the net polarization vector of the resultant films to orient in the 3 direction.

7. For Fig 1a, the authors have demonstrated a comparison between the graphene substrate and $Ti_3C_2T_x$ substrate. A brief explanation in the particular reason for graphene as a reference should be added. Then, the authors discussed the microscopic force between the first layer and the second layer. However, more explanations should be carried out according to the surface energy and bond interactions. As more chains fill the free space on the $Ti_3C_2T_x$ substrate, the microstructure will change between the first and second layers. Correspondingly, surface relaxation and reconstruction etc. may also happen. A more detailed discussion about morphology and interactions should be included in order to reach the result of electrostatic force, bond conformations, and electroactive phase.

Graphene was chosen as a comparative system for two main reasons: (1) it has a 2D geometry and is the most well studied 2D material in literature; and (2) graphene, when locked into a planar configuration (i.e., it cannot roll up or be deformed by the polymer film), is not polarizable and does not have a strong electrostatic field above and below the plane. We have added this reasoning into the main text:

“As an atomically thin 2D sheet, graphene does not exhibit out of plane polarizability and is thus a suitable comparative system to $Ti_3C_2T_x$.¹⁹”

We have thoroughly discussed the influence of the substrate interactions on the PVDF-TrFE co-polymer bond conformations, and consequently the electroactive phase fractions, in the Supplementary Information. The discussion is inserted below for convenience:

“To understand the phase distributions of the PVDF-TrFE co-polymer film adjacent to the $Ti_3C_2T_x$ nanosheet substrate, the probability distributions of the dihedral angles were

monitored for a 14-chain PVDF-TrFE co-polymer film (Fig. S7). The PVDF-TrFE co-polymer consists of three commonly found phases, namely the α phase (non-polar), γ phase (semi-polar) and the β phase (highly polar).⁶ These phases correspond to spatial conformation of the bonds, either *trans* (T) or *gauche* (G). The α phase is thermodynamically favored in fluoropolymers, due to its *trans-gauche* (TGTG'TGTG') conformation, which consists of 50% *trans* bonds and 50% *gauche* bonds.⁴ Conversely, the β phase is an all-*trans* (TTTTTTTT) conformation, which spatially separates the H moieties on one C atom from the F atoms on the adjacent C atom, giving rise to a strong H-F dipole moment.⁶ The γ phase is a stable intermediate state between the α phase and the β phase, as evidenced by the 75% *trans* and 25% *gauche* fraction (TTTGTTTG'), giving rise to dipole moments which result in a lower maximum polarization relative to the β phase.⁴ Hence, the distribution of the dihedral angles and subsequently the phase fractions can provide insight on the changes in local electroactivity of the PVDF-TrFE co-polymer film adjacent to the Ti₃C₂T_x nanosheet substrate.

The distribution of the dihedral angles (Fig. S7a) was taken as the average of all angles in a 14-chain PVDF-TrFE co-polymer film interacting with an immobile Ti₃C₂T_x nanosheet substrate over the span of the simulation (4 ns). The PVDF-TrFE co-polymer chains exhibited configurations at four main dihedral angles, $\pm 180^\circ$ (*trans*) and $\pm 60^\circ$ (*gauche*).⁶ As a function of simulation time, the fraction of *trans* conformation (Fig. S7b, black line) was found to increase and attain a final value of approximately 63%, whereas approximately 37% of the bonds were observed in the *gauche* conformation (Fig. S7, red line). These values correspond to either a majority of α phase (74%) with low prevalence of the β phase (26%), or a near-even distribution of the α phase (48%) and γ phase (52%), or a combination of the two. Importantly, while the PVDF-TrFE generally crystallizes into the β phase due to the third F atom in the TrFE monomer, these values suggest a large presence of the α phase.⁶ Indeed, at the local level, the experimental data observed the presence of the α and γ phases adjacent to the Ti₃C₂T_x nanosheet (Fig. 3d, e); however, the FTIR (Fig. S16) and XRD (Fig. S17, Fig. S18) data presented further in this document suggest the β phase as the primary conformation in the bulk of the Ti₃C₂T_x/PVDF-TrFE composites.

Similarly, the temporal evolution of the dihedral angles was repeated for a 70-chain PVDF-TrFE co-polymer film on the Ti₃C₂T_x nanosheet or graphene substrate (Fig. S8). Similar to the 14-chain PVDF-TrFE co-polymer films (Fig. S7b), the larger films on a Ti₃C₂T_x nanosheet substrate exhibited a larger *trans* fraction (approximately 65%) relative to the *gauche* fraction (approximately 35%). Interestingly, when simulated adjacent to a graphene substrate, the same 70-chain PVDF-TrFE copolymer film exhibited a lower fraction of *trans* bonds (approximately 57%) and subsequently a higher fraction of *gauche* bonds (approximately 43%).”

The Ti₃C₂T_x possesses a negative surface charge (with O charge of $q_O = -0.24e$) which interacts with the positive CH₂ charge of the PVDF-TrFE (with C and H charges of $q_C = +0.48e$ and $q_H = +0.06e$, respectively). Such electrostatic interactions have been demonstrated by MD simulations to induce the β phase of PVDF chains during interactions with TiO₂ nanoparticles.²² Here, an increase in the fraction of *trans* conformations is observed in the

PVDF-TrFE/Ti₃C₂T_x as a function of simulation time (Supplementary Information Figure S6), however the maximum observed fraction of *trans* conformations is 63%, corresponding to either a majority of α phase (74%) with a low prevalence of β phase (26%), a near-even distribution of α phase (48%) and γ phase (52%), or a combination of the two. The experiments presented in this manuscript show that the fraction of electroactive phases ($\beta+\gamma$) adjacent to the Ti₃C₂T_x nanosheets decreased (Main Text Figure 3d). The structure of the co-polymer chains can strongly affect the dipole moment, as shown by density functional theory calculations.²³

In addition, the microstructure of the co-polymer film between the different layers is quantified by the evolution of the film local density as function of the distance from the substrate. The **cumulative** density of the co-polymer film on the Ti₃C₂T_x and graphene substrates is replaced by the **local** film density in the Supplementary Information (Supplementary Information Figure S5), shown in Figure R7 below for convenience. The local density of the first layer of copolymers adjacent to the graphene and Ti₃C₂T_x surface is approximately 2.3 g cm⁻³ and 1.6 g cm⁻³, respectively. The larger local density of the co-polymer film adjacent to the graphene substrate indicates that the PVDF-TrFE chains are more packed than those adjacent to the Ti₃C₂T_x, as the latter has a rougher surface imposed by the OH groups that induce steric effects in the co-polymer chains. The local density of the layers further away from the substrate is practically identical in both Ti₃C₂T_x/PVDF-TrFE and graphene/PVDF-TrFE systems.

The above discussion about the co-polymer local morphology is included in the Supplementary Information of the revised manuscript.

Fig. R7: Distribution of the local film density of the PVDF-TrFE co-polymer film (14 chains) as a function of the distance from the Ti₃C₂T_x nanosheet (red line) and the graphene sheet (black line).

8. For the mechanical property part of the SEA extrusion printed Ti₃C₂T_x/PVDF-TrFE, the authors have demonstrated the ductility from yield point and material elongations. A

homogenous dispersion result is reached due to the linear increase in Young's modulus. However, since the slope of the stress-strain curve is proportional to elastic modulus and elastic modulus is also proportional to the slope of force curve dF/dr at equilibrium distance, a more detailed explanation between homogenous dispersion and bond strength should be included based on the result of elastic modulus. Moreover, as a material is elastically elongated under a tensile load, there is a change in Poisson's ratio. An all-rounded discussion corresponding to lateral and longitudinal is suggested.

The authors agree with the reviewer's statement that a homogenous dispersion has been reached between the $Ti_3C_2T_x$ nanosheet and the PVDF-TrFE molecules. While, as stated by the reviewer, the change in Young's modulus can present a tool to probe the bond strength at the interface of the PVDF-TrFE/ $Ti_3C_2T_x$ interface this assumes an ideal system of load transfer. The composite materials described in the manuscript are homogenous dispersions on the micro/nano level, however macroscopically the SEA extrusion printing technique leads to variations in thickness arising from the printing patterns of extrusion (see Figure R4). These variations in thickness of approximately 2 μm lead to local areas of stress concentration and obfuscate the true bond strength between the polymer and the $Ti_3C_2T_x$ nanosheet.

In order to understand this force interaction, MD simulations were performed to determine the 'threshold' force that a single polymer chain could be removed from the $Ti_3C_2T_x$ nanosheet which corresponded to 4.17 pN of force (compared to 2.78 pN for a graphene surface).

We have provided a discussion on page 5 of the Main Text for the $Ti_3C_2T_x$ /PVDF-TrFE interactions:

“These simulations revealed an extremely strong electrostatic interaction between the PVDF-TrFE chains and the $Ti_3C_2T_x$ nanosheet, requiring ~ 4.17 pN of force to desorb one PVDF-TrFE chain from the $Ti_3C_2T_x$.”

and on page 7 of the Main Text for the graphene/PVDF-TrFE interactions:

“In contrast to the fluoropolymer film on $Ti_3C_2T_x$ nanosheet, the fluoropolymer film on graphene is able to migrate on the periodic lattice easily and requires a significantly lower force (~2.78 pN) to detach a single fluoropolymer chain from the lattice, indicating a weaker interaction between the components.”

In the authors' opinion, an in-depth study of the mechanical properties of the $Ti_3C_2T_x$ /PVDF-TrFE composite is beyond the scope and novelty of the paper.

9. The authors have discussed the different PVDF-TrFE phases which is determinant in crystallinity and energy harvesting capabilities. If possible, a microscopic morphology schematic of each phase and compositions is helpful to demonstrate the result and relationships.

We thank the reviewer for this comment. We have added a simplified schematic of the process in the Main Text and in Figure R6, comparing the polarization alignment of randomly oriented (Figure R6a) β phase PVDF-TrFE chains through the currently well-accepted electrical poling process (Figure R6b,c) to the polarization locking method (Figure R6d,e). Please see the discussion in the response to comment 6 above.

10. The whole manuscript is based on fluoropolymer; however, the authors haven't mentioned the effect of temperature (e.g., temperature-dependent relaxations on the surface between Nanosheet and bulk copolymer, polymer temperature-dependent phases, temperature influence on trans or gauche compositions, etc.). Discussing the influence of temperature would be valuable for the manuscript.

We thank the reviewer for their suggestion. We have undertaken differential scanning calorimetry (DSC) in order to investigate the temperature dependent properties and discussed the results in the Supplementary Information. The DSC thermograms are presented in Figure R8 below for convenience. The PVDF-TrFE co-polymer contains two peaks of interest in the DSC thermograms, namely the Curie temperature (T_c) between 105 °C and 110 °C, and the melting temperature (T_m) between 140 °C and 145 °C. The T_c represents a relaxation of the PVDF-TrFE co-polymer chains from a high energy (ferroelectric) dipole moment state to a low energy (paraelectric) state, losing the piezoelectric properties in the process. Hence, based on the data obtained, the $Ti_3C_2T_x$ /PVDF-TrFE films retain their energy conversion properties up to approximately 105 °C. It is widely accepted in literature that the d_{33} of piezoelectric and ferroelectric materials decreases to $d_{33} \approx 0$ pC N⁻¹ at temperatures above the T_c .

Fig. R8: DSC thermograms of the SEA extrusion printed $Ti_3C_2T_x$ /PVDF-TrFE films, offset for clarity.

Reviewer #3 (Remarks to the Author):

I reviewed the paper by Shepelin et al. on the piezoelectric polarisation formed between PVDF-

TrFE and Ti₃C₂T_x. The paper is interesting, shows some timely results involving simulations and measurements, and I believe that it may have an impact in the field. I'd have few questions as shown below:

We thank the reviewer for their detailed reading of the manuscript.

1) According to the data, one of the main driving forces for the stabilisation of the polarisation vector perpendicular to the surface is the strong electrostatic interactions between the macro-molecule and Ti₃C₂T_x. However, I have not seen any simple charge density plots that would show clearly this effect. I also checked the SI but nothing there. I'd suggest the author to show a few charge density difference plots which will help to set down the idea of the strong electrostatic interactions. At the moment it looks indirect though. We can see the effect in terms of the polarisation vector but we don't see the main players creating it.

We thank the reviewer for the suggestion. The electrostatic interactions between the Ti₃C₂T_x surface and the copolymer film induce structural changes in the PVDF-TrFE chains, which are more elongated and richer in trans conformations than those in the PVDF-TrFE/graphene system where the substrate has zero charge. This is consistent with the literature,^{22,23} where differences in the dipole moment are reported based on the polymer structure (e.g., between the β and α phase of PVDF, associated with the trans and gauche conformations). It should be noted here that these differences are attributed directly to the substrate effect in vacuum at 25 °C, however in practice the effects of solution and polymer crystallization kinetics will also play a role in trans and gauche confirmation. Crucially, the MD simulations show the locked dipole orientation of these polymer chains is induced by the Ti₃C₂T_x substrate.

Furthermore, the charge density maps of the copolymer film have been calculated as function of the distance from the Ti₃C₂T_x (left) and graphene surface (right), as shown in Figure R9 below:

Fig. R9: The spatial charge distribution of the PVDF-TrFE copolymer film (70 chains) on the **a-c** $\text{Ti}_3\text{C}_2\text{T}_x$ nanosheet and **d-f** graphene at distances of 5 Å (left), 15 Å (middle) and 25 Å (right) from the substrate.

No significant change was observed in the charge distributions as the thickness of the copolymer film increased.

The local charge density of the copolymer film is shown in Figure R10 below as function of the distance from the $\text{Ti}_3\text{C}_2\text{T}_x$ (red line) and graphene surface (black line), averaged over time. The local charge density follows qualitatively the same evolution with the local film density (please see response to Reviewer 2, Comment 7). Specifically, the first layer of the copolymer film exhibits a lower charge density when $\text{Ti}_3\text{C}_2\text{T}_x$ is used as substrate, as the chains adjacent to the $\text{Ti}_3\text{C}_2\text{T}_x$ surface are less packed ($\text{Ti}_3\text{C}_2\text{T}_x$ is rougher than graphene imposing steric effects). However, the local charge densities both in $\text{Ti}_3\text{C}_2\text{T}_x/\text{PVDF-TrFE}$ and graphene/ PVDF-TrFE are qualitatively the same. Therefore, no correlation between the charge density and the polarization vector is observed.

Fig. R10: The local charge density of the PVDF-TrFE copolymer film as a function of the distance from the substrate for the $\text{Ti}_3\text{C}_2\text{T}_x$ nanosheet (red line) and graphene (black line), averaged over time.

2) On page 7, it is mentioned that the electrostatic screening length in 2D materials is 1-10 nm, no citations or references are included whatsoever. Moreover, there is some speculation on the origin of the polarisation locking via electrostatic forces, which is related with comment 1) above, but not solid data is shown. I'd suggest the author to undertake a careful analysis on the clarification of the origin of the locking effect before stating anything else.

We thank the reviewer for their comments and suggestions. The citation is present in the main text of the manuscript and we have updated the location of the citation in question (reference 31) for clarity. The updated text is presented below for convenience:

“Given that the electrostatic screening length in 2D materials is between 1 nm and 10 nm,³¹ and considering the strength of the electrostatic interaction observed between the PVDF-TrFE co-polymer and the $\text{Ti}_3\text{C}_2\text{T}_x$ nanosheet, it is highly probable that this polarization locking is occurring as a consequence of electrostatic forces.¹⁹”

As discussed in response to Comment 7 from Reviewer 2, $\text{Ti}_3\text{C}_2\text{T}_x$ has a negative surface charge (with O charge of $q_o = -0.24e$) which interacts with the positive CH_2 charge of the PVDF-TrFE (with C and H charges of $q_c = +0.48e$ and $q_h = +0.06e$, respectively). Such electrostatic interactions induce the f_3 phase of the PVDF chains, as shown for PVDF polymers interacting with TiO_2 nanoparticles by MD simulations.²² Here, an increased fraction of *trans* conformations is observed in the PVDF-TrFE/ $\text{Ti}_3\text{C}_2\text{T}_x$ (Supplementary Information: Fig. S6), which are associated with the f_3 -PVDF phase. The structure of the co-polymer chains can affect strongly the dipole moment, as shown by density functional theory calculations.²³

3) It has been show recently by Tian et al. Nano Letters 2020, 20, 2, 841–851 that the electronic polarisability is the fundamental variable for the dielectric properties of 2D materials. If

electrostatic interactions are related to the locking phenomena, then I'd assume that the strong anisotropy between both kind of interactions (in-plane, out-of-plane) will play a role in the effect. This is also related with the short interactions $\sim 5 \text{ \AA}$ perpendicular to the layer mentioned by the authors (page 7), but not fully referenced. I would suggest the authors to calculate the range of interactions involved in the interactions between $\text{Ti}_3\text{C}_2\text{T}_x$ and the macro-molecule using the methods in Tian et al. This will give a good idea about the length where the interaction dying out in the system.

We thank the reviewer for the recommendation. The methodology proposed in Tian *et al.*²⁵ relies on the dielectric tensor which was obtained by first principle calculations. The present classic Molecular Dynamics simulations do not allow the calculation of variables obtained by density functional theory. The non-bonded interactions (van der Waals and Coulombic forces) in the present simulations are obtained based on the Transferable, Extensible, Accurate and Modular (TEAM-FF) forcefield. We have calculated the van der Waals and Coulombic forces exerted by $\text{Ti}_3\text{C}_2\text{T}_x$ on various layers (5, 10, 15 and 20 \AA) of the copolymer film and their temporal evolution is shown in Figure R11 below. The Coulombic forces (red lines) exerted by the $\text{Ti}_3\text{C}_2\text{T}_x$ substrate to the first layer of the PVDF-TrFE co-polymer (within 5 \AA from the substrate) in the direction perpendicular to the substrate are $-1747.4 \text{ kcal mol}^{-1} \text{ \AA}^{-1}$ and become much weaker in the next layers further away from the substrate (of the order of $-300 \text{ kcal mol}^{-1} \text{ \AA}^{-1}$). The Lennard-Jones forces double from the first to the second layer but do not alter for distances larger than 12 \AA , which was used as cutoff distance for the Lennard-Jones interactions in the present simulations.

Fig. R11: The temporal evolution of the van der Waals forces (grey line) and the Coulombic forces (red line) exerted by the $\text{Ti}_3\text{C}_2\text{T}_x$ nanosheet on the PVDF-TrFE copolymer film, measured at distances from the $\text{Ti}_3\text{C}_2\text{T}_x$ nanosheet of 5 \AA (top left), 10 \AA (top right), 15 \AA (bottom left) and 20 \AA (bottom right).

4) On page 8, it is also mentioned the local polarisation but no data included to unveil its origin. Please improve.

Comparisons of the co-polymer films for $\text{Ti}_3\text{C}_2\text{T}_x$ and graphene show hardly any difference in their charge distributions (please refer to the response for Comment 1, Reviewer 3, Fig. R10), indicating that locking of the polarization could be associated to the molecule structure (i.e., fraction of *trans* and *gauche* conformations, or elongated vs. coiled structures) which, in turn, is affected by the electrostatic interactions between $\text{Ti}_3\text{C}_2\text{T}_x$ and the co-polymer film. This is also supported by the calculations of the dipole moment vectors of individual co-polymer chains from the film, which exhibit a correlation between their structure and their corresponding dipole moment vector. Figure R12 shows exemplarily a snapshot of five copolymer chains on $\text{Ti}_3\text{C}_2\text{T}_x$ and graphene substrates along with their individual dipole moment vectors. The rest of the copolymer chains (65 chains out of 70) are omitted for clarity. The chains on the $\text{Ti}_3\text{C}_2\text{T}_x$ substrate are clearly more elongated than those on top of graphene, attaining individual dipole moment vectors perpendicular to the basal plane of $\text{Ti}_3\text{C}_2\text{T}_x$, similar to the total dipole moment vector of the 70-chain film. Contrary to the copolymer chains on the $\text{Ti}_3\text{C}_2\text{T}_x$ substrate, those on top of graphene are more coiled and the individual dipole moment vectors attain random orientations. We have added the temporal evolution of the dipole moment vectors of individual copolymer chains to the manuscript as Supplementary Video S2.

Fig. R12: Snapshots of five PVDF-TrFE copolymer chains on **a** the $\text{Ti}_3\text{C}_2\text{T}_x$ substrate and **b** the graphene substrate, along with their individual polarization vectors.

5) Can the authors show some histogram of the amount of devices fabricated and their performance? This is very important to proof the accuracy of the device-data, and not just one device worked out fine.

We thank the reviewer for their comment. The PFM measurements were undertaken on one, randomly selected film at each $\text{Ti}_3\text{C}_2\text{T}_x$ concentration, whereby the location of the PFM

measurement on the film was selected approximately 1 cm from the edge in order to reduce thickness variability between samples. Notably, the contribution from thickness changes has been calculated as minimal *via* in-depth discussion in the Supplementary Information. Furthermore, one cantilever was used to investigate all samples presented in this manuscript, thus the contribution from variation in cantilever characteristics was eliminated. For each applied voltage value, a rectangular area with dimensions at 0.195 μm width (10 px) and 3.906 μm height (200 px) was measured. In total, this corresponds to 2000 data points, with the separation between pixels approximately equal to the cantilever tip radius at 19.5 nm. The operation to obtain the $A\cos(\varphi)/Q_f$ was undertaken on a per-pixel basis prior to obtaining an average over the measured area to preserve the effective d_{33} value for each individual pixel.

Here, the trends for the improvements in the effective d_{33} can be considered universal, considering the tremendously low error within the samples. Furthermore, the observed $\text{Ti}_3\text{C}_2\text{T}_x$ nanosheet-induced polarization locking mechanism was observed to occur in solution prior to the SEA extrusion printing method, therefore both the SEA extrusion printed 0.50 wt% $\text{Ti}_3\text{C}_2\text{T}_x$ /PVDF-TrFE film and the solvent cast 0.50 wt% $\text{Ti}_3\text{C}_2\text{T}_x$ /PVDF-TrFE film were expected to exhibit similar effective d_{33} values. Indeed, the difference in the effective d_{33} values between the two samples was 0.001 pm V^{-1} , well within error (0.038 pm V^{-1} for printed and 0.078 pm V^{-1} for solvent cast). We are therefore confident in the enhancement effect observed *via* PFM.

In order to confirm the polarization locking effect at the macroscale, a SEA extrusion printed pristine PVDF-TrFE copolymer film and a SEA extrusion printed 0.50 wt% $\text{Ti}_3\text{C}_2\text{T}_x$ film were chosen, based on the results of the PFM investigation. In this experiment, we have tested between five and six energy harvesting devices at each $\text{Ti}_3\text{C}_2\text{T}_x$ concentration, with sputter coated Cr/Au electrodes to collect the generated charge and wires soldered to Cu foil adhesive to transfer the charge to the measuring instrument, with the detailed method in the Supplementary Information. Each energy harvesting device was exposed to 60 compression cycles and the d_{33} was calculated by $d_{33} = D_3/\sigma_3 (Q/A_e)/(F/A_\sigma)$ for each individual compression cycle. The resultant data is presented in Figure R13 below. The reason for using the best-performing sample for each was three-fold: (1) the acrylic-based adhesive of the Cu foil is pressure sensitive,²⁶ with particulate conductive additives acting to form conductive networks between the Cu foil and the underlying Cr/Au electrode,²⁷ therefore notably the decrease in generated charge can be attributed to a lack of conductive pathways under minimal pressure (capacitive coupling);¹⁹ and (2) following the discussion in the Supplementary Information, the minimization of the contribution from triboelectricity, flexoelectricity and stray electromagnetic interference ensures the resultant measured charge arises only due to piezoelectricity. In this experiment, the volume between the load cell of the mechanical tester was shielded from electromagnetic interference and the load cell was grounded to ensure no contribution arose from charge transfer through the load cell. Furthermore, we have ensured that no contribution from triboelectricity was present by adhering each component of the energy harvesting device to the SEA extrusion printed $\text{Ti}_3\text{C}_2\text{T}_x$ film. We have furthermore tested the dependence of the generated charge on the applied stress (Supplementary Information Figure S29), observing a linear relationship and thus confirming the sole presence

of piezoelectric contribution. Therefore, we believe the highest performing devices exhibit true piezoelectric energy conversion, with the lower performing devices demonstrating a lower charge generation due to capacitive coupling rather than surface charge.

Fig. R13: The measured piezoelectric charge coefficient (d_{33}) values for a minimum of five SEA extrusion printed $Ti_3C_2Tx/PVDF-TrFE$ energy harvesters per Ti_3C_2Tx concentration, with each data point corresponding to a single compression cycle. Each individual energy harvester was subjected to 60 compression cycles.

6) Similar to 5), how many samples were used to extract the value of d_{33} ? Some histogram would be helpful to have some statistics of its magnitude.

We thank the reviewer for their comment and kindly ask to refer to the response to comment 5 above.

7) The ink part is very interesting, but I missed some discussions whether other solvents were tried or utilised in the process of fabrication the devices.

We thank the reviewer for their comment and finding merit in the ink aspect. In this manuscript, we have used acetone as the only solvent. Prior work, both within our group and elsewhere, has focused primarily on the manufacture of fluoropolymer films using *N,N*-dimethylformamide (DMF), dimethyl sulfoxide (DMSO), *N*-methyl-2-pyrrolidone (NMP) or *N,N*-dimethylacetamide (DMAc) in significant quantities (>30% of the total solvent volume fraction). It has been widely considered that these solvents are most suitable for dissolving fluoropolymers; however, the drawbacks of such solvents are the high boiling point (>150 °C at 101.3 kPa),²⁸⁻³¹ low vapor pressure (<0.5 kPa at 21 °C)³² and high toxicity,²⁸⁻³¹ all of which are detrimental for the high-speed and large-scale production of electromechanical energy conversion devices. In one of our previous works,¹⁷ we have investigated the volume ratio of DMF:acetone via the Hansen solubility parameters, finding the optimal ratio at approximately 50 vol% DMF. More recently, we have discovered the recycling capability of PVDF-TrFE directly in acetone.²⁰ In comparison to the aforementioned solvents, acetone possesses a

significantly lower boiling point (56 °C at 101.3 kPa),³³ high vapor pressure (26 kPa at 21 °C)³² and reduced toxicity, reported as one of the least toxic industrial solvents.³³ We have updated our rationale on the elimination of DMF, which can be found in the Supplementary Information and is shown below for convenience:

“Recently, we described the dissolution and recycling of SEA extrusion printed PVDF-TrFE co-polymer films using acetone as the only solvent. Here, N,N-dimethylformamide (DMF) was completely eliminated as a solvent for extrusion printing entirely and replaced by acetone. Acetone has inherent advantages, with faster evaporation rates that enable rapid crystallization and drying of SEA extrusion printed polymer films.” In particular, DMF exhibits a high boiling point (>150 °C at 101.3 kPa),² low vapor pressure (<0.5 kPa at 21 °C)² and high toxicity.² Comparatively, acetone exhibits a low boiling point (56 °C at 101.3 kPa),⁴ high vapor pressure (26 kPa at 21 °C)³ and reduced toxicity, reported as one of the least toxic industrial solvents,⁴ and is thus better suited for SEA extrusion printing.”

Furthermore, we have undertaken extensive rheological characterization of the PVDF-TrFE/acetone ink, finding exceptional shear thinning properties of this ink relative to the previously utilized DMF:acetone (40:60 vol%), which are important for 3D printing applications. The characterization is shown in Supplementary Information Figure S10a and presented in Figure R14a below for convenience, and the in-depth discussion is present in the Supplementary Information:

Fig. R14: Steady-state rheology. **a** The shear rate sweeps for inks of PVDF-TrFE/(DMF:acetone, 40:60 vol%) and PVDF-TrFE/acetone, containing PVDF-TrFE (35 wt%). **b** The applied shear rate profile as a function of time and **c** the resultant viscosity for PVDF-TrFE/acetone inks with PVDF-TrFE concentration at 35 wt% and 45 wt%.

Additionally, the oscillatory rheology results shown in Supplementary Information Figure S11 exemplify a strong interaction between the PVDF-TrFE and acetone, whereby the ink behaves like a physical gel, presented in Figure R15 below for convenience:

Fig. R15: Oscillatory rheology measurements on PVDF-TrFE/acetone inks. **a-c** The frequency sweeps for **a** 35 wt%, **b** 40 wt% and **c** 45 wt% PVDF-TrFE/acetone inks, obtained at fixed strain ($\gamma_s = 1\%$). **d-f** The shear strain sweeps for **d** 35 wt%, **e**

40 wt% and **f** 45 wt% PVDF-TrFE/acetone inks, obtained at fixed frequency ($\omega = 1$ Hz).

8) Regarding degradation, are the films stable under radiation e.g. light, laser power, etc. Some comments/data regarding external driving forces used to characterise the samples would be helpful for further verification from other groups.

We thank the reviewer for their comment. The $\text{Ti}_3\text{C}_2\text{T}_x/\text{PVDF-TrFE}$ inks were visually observed to be stable without significant aggregation of the $\text{Ti}_3\text{C}_2\text{T}_x$ nanosheets over a five month timespan, as demonstrated in Supplementary Information Figure S3 (compared to single-walled carbon nanotube/PVDF-TrFE inks). Colloidal solutions of $\text{Ti}_3\text{C}_2\text{T}_x$ have been reported to oxidize to TiO_2 over time when stored at ambient conditions,³⁴ however this manuscript used acetone as the solvent and thus the $\text{Ti}_3\text{C}_2\text{T}_x$ nanosheets were not expected to oxidize. Moreover, the PVDF-TrFE co-polymer is known to be an exceptionally chemically stable polymer. Its transparent nature (Supplementary Information Figure S14) means that it typically absorbs very little energy from a photon and hence it remains stable over the timeframes we have studied. The addition of $\text{Ti}_3\text{C}_2\text{T}_x$ nanosheets to the PVDF-TrFE copolymer can lead to laser-induced degradation due to localized heating adjacent to individual $\text{Ti}_3\text{C}_2\text{T}_x$ nanosheets (we have observed similar effects in our previous work using single-walled carbon nanotubes). To this end, the laser power and exposure on the sample during the Raman microscopy analysis was minimized and repeated mapping was performed to ensure no degradation of the films was induced by our characterization conditions. Specific studies on the laser-induced degradation of the $\text{Ti}_3\text{C}_2\text{T}_x/\text{PVDF-TrFE}$ composites were not performed. It should be noted that for practical applications, the samples will be coated with conductive (opaque) electrodes that will typically prevent any exposure to light.

References

1. IEEE Standard on Piezoelectricity, 1987 (ANSI/IEEE Std. 176-1987) (1988).
2. New developments in composites, copolymer technologies and processing techniques for flexible fluoropolymer piezoelectric generators for efficient energy harvesting. *Energy Environ. Sci.* **12**, 1143–1176 (2019).
3. Furukawa, T. Structure and functional properties of ferroelectric polymers. *Adv. Colloid Interface Sci.* **71-72**, 183–208 (1997).
4. van den Ende, D. A., de Almeida, P. & van der Zwaag, S. Piezoelectric and mechanical properties of novel composites of PZT and a liquid crystalline thermosetting resin. *J. Mater. Sci.* **42**, 6417–6425.
5. Lee, B. Y. *et al.* Virus-based piezoelectric energy generation. *Nat. Nanotechnol.* **7**, 351–356 (2012).
6. Deutz, D. B. *et al.* Analysis and experimental validation of the figure of merit for piezoelectric energy harvesters. *Mater. Horiz.* **5**, 444–453 (2018).
7. Thakur, P. *et al.* Superior performances of *in situ* synthesized ZnO/PVDF thin film based self-poled piezoelectric nanogenerators and self-charged photo-power bank with high durability. *Nano Energy* **44**, 456-467 (2018).
8. Zhao, Q. *et al.* Flexible textured MnO₂ nanorods/PVDF hybrid films with superior piezoelectric performance for energy harvesting application. *Compos. Sci. Technol.* **199**, 108330 (2020).
9. Kumar, A., Kumar, A. & Prasad, K. Power generation characteristics of 0.50(Ba_{0.7}Ca_{0.3})TiO₃-0.50Ba(Zr_{0.2}Ti_{0.8})O₃/PVDF nanocomposites under impact loading. *J. Mater. Sci. Mater. Electron.* **31**, 12708-12714 (2020).
10. Zhou, Z. *et al.* Enhanced piezoelectric and acoustic performances of poly(vinylidene fluoride-trifluoroethylene) films for hydroacoustic applications. *Phys. Chem. Chem. Phys.* **22**, 5711-5722 (2020).
11. Yang, L. *et al.* Effect of rolling temperature on the microstructure and electric properties of β -polyvinylidene fluoride films. *J. Mater. Sci. Mater. Electron.* **29**, 15957-15965 (2018).
12. Yang, L., *et al.* Enhanced electrical properties of multiwalled carbon nanotube/poly(vinylidene fluoride) films through a rolling process. *J. Mater. Sci. Mater. Electron.* **25**, 2126-2137 (2014).
13. Sousa, R. E., *et al.* Microstructural variations of poly(vinylidene fluoride co-hexafluoropropylene) and their influence on the thermal, dielectric and piezoelectric properties. *Polym. Test.* **40**, 245-255 (2014).
14. Xia, W., *et al.* Dielectric, piezoelectric and ferroelectric properties of a poly(vinylidene fluoride-co-trifluoroethylene) synthesized via a hydrogenation process. *Polymer* **54**, 440-446 (2013).
15. Vacche, S. D., *et al.* The effect of processing conditions on the morphology, thermomechanical, dielectric and piezoelectric properties of P(VDF-TrFE)/BaTiO₃ composites. *J. Mater. Sci.* **47**, 4763-4774 (2012).

16. TA Instruments, 5500 Brochure for General Materials (2015). Accessed on 13/10/2020. Available at: https://www.tainstruments.com/wp-content/uploads/5500_Access_EM_NoPrice_082615_low.pdf
17. Shepelin, N. A. *et al.* 3D printing of poly(vinylidene fluoride-trifluoroethylene): a poling-free technique to manufacture flexible and transparent piezoelectric generators. *MRS Commun.* **9**, 159-164 (2019).
18. Bailey, T. & Hubbard, J. E. Jr. Distributed piezoelectric-polymer active vibration control of a cantilever beam. *J. Guid. Control Dyn.* **8**, 605-611 (1985).
19. Measurement Specialties Inc. *Piezo Film Sensors Technical Manual*, Norristown, PA (1999).
20. Shepelin, N. A. *et al.* Printed recyclable and self-poled polymer piezoelectric generators through single-walled carbon nanotube templating. *Energy Environ. Sci.* **13**, 868-883.
21. Kumar, P. *et al.* Ultrahigh electrically and thermally conductive self-aligned graphene/polymer composites using large-area reduced graphene oxides. *Carbon* **101**, 120-128 (2016).
22. Hong, H., Song, S. A. & Kim, S. S. Phase transformation of poly(vinylidene fluoride)/TiO₂ nanocomposite film prepared by microwave-assisted solvent evaporation: An experimental and molecular dynamics study. *Compos. Sci. Technol.* **199**, 108375 (2020).
23. Wang, W., Fan, H. & Ye, Y. Effect of electric field on the structure and piezoelectric properties of poly(vinylidene fluoride) studied by density functional theory. *Polymer* **51**, 3575-3581 (2010).
24. Claude, J. *et al.* Electrical Storage in Poly(vinylidene fluoride) based Ferroelectric Polymers: Correlating Polymer Structure to Electrical Breakdown Strength. *Chem. Mater.* **20**, 2078-2080 (2008).
25. Tian, T., *et al.* Electronic Polarizability as the Fundamental Variable in the Dielectric Properties of Two-Dimensional Materials. *Nano Lett.* **20**, 841-851 (2020).
26. 3M, 3M Conductive Copper Foil Tape 1181 Data Sheet (2011). Accessed on 12/11/2020. Available at: <https://multimedia.3m.com/mws/media/373700/3m-copper-foil-tape-1181-with-conductive-adhesive.pdf>
27. Ghosh, D., Mccutcheon, J. W. & Culler, S. R. An electrically conductive article containing shaped particles and methods of making same. World patent WO2015073346A1 (2014).
28. Bipp, H. & Kieczka, H. Formamides. in *Ullman's Encyclopedia of Industrial Chemistry* (Wiley-VCH, Weinheim, 2011).
29. Roy, K. M. Sulfones and Sulfoxides. in *Ullmann's Encyclopedia of Industrial Chemistry* (Wiley-VCH, Weinheim, 2000).
30. Harreus, A. L. *et al.* 2-Pyrrolidone. in *Ullmann's Encyclopedia of Industrial Chemistry* (Wiley-VCH, Weinheim, 2011).
31. Cheung, H., Tanke, R. S. & Torrence, G. P. Acetic Acid. in *Ullmann's Encyclopedia of Industrial Chemistry* (Wiley-VCH, Weinheim, 2011).
32. Smallwood, I. M. *Handbook of organic solvent properties* (John Wiley & Sons, New York, 1996).

33. Sifniades, S. & Levy, A. B. Acetone. in *Ullmann's Encyclopedia of Industrial Chemistry* (Wiley-VCH, Weinheim, 2000).
34. Zhang, C. J. *et al.* Oxidation Stability of Colloidal Two-Dimensional Titanium Carbides (MXenes). *Chem. Mater.* **29**, 4848-4856 (2017).

REVIEWER COMMENTS

Reviewer #1 (Remarks to the Author):

I am happy with the corrections and response to comments. Berlincourt has been used for piezo d_{33} coefficients. Good response to other reviewer comments.

C_Bowen

Reviewer #2 (Remarks to the Author):

After revision, the paper reads more reasonable. The additional discussions in polarization along Z3(d_{31} and d_{33}) are more comprehensive. To further improve the manuscript quality, the authors are suggested to address the following questions:

1. The interconnection of d_{31} and d_{33} doesn't matter. For the d_{31} measurement, the authors may test the strain – electric field relationship, or indirectly deduce d_{31} from d_{33} . For piezoelectric materials, d_{31} is also a critical parameter, the authors should at least provide an approximate value.
2. There are some other papers (See "High-performance micromachined unimorph actuators based on electrostrictive poly (vinylidene fluoride–trifluoroethylene) copolymer" and "Insect-scale fast moving and ultrarobust soft robot") introducing the PVDF or PVDF-TrFE actuators. For an actuator, the authors can coat some other elastic material to the PVDF-TrFE film or fabricate an actuator by laminating method.
3. The demonstration of three-phase PVDF-TrFE copolymer with dihedral angle and trans/gauge-state is nice. The authors have demonstrated the α and γ phases are adjacent to Ti_3C_2Tx where β phase is the main conformation in the bulk composite. Is there a quasi-morphotropic boundary or will there be a micro/elemental distortion in the bulk composites? Is there a change in the net polarization or Curie Point for the nanosheet based on the distribution and phase composition or fraction?

Reviewer #3 (Remarks to the Author):

The authors have addressed all my concerns and questions. The improved version of the manuscripts looks pretty solid. I don't have any additional comment. The paper can be accepted.

DETAILED RESPONSE TO REVIEWERS

The authors would like to thank the editors for their time, expertise, and dedication in reviewing our manuscript, please find below a detailed point-by-point response to the reviewers' comments and recommendations.

Reviewer #1 (Remarks to the Author):

I am happy with the corrections and response to comments. Berlicourt has been used for piezo d33 coefficients. Good response to other reviewer comments.

C_Bowen

We thank the reviewer for their comment and are pleased to hear the positive feedback.

Reviewer #2 (Remarks to the Author):

After revision, the paper reads more reasonable. The additional discussions in polarization along Z3(d31 and d33) are more comprehensive. To further improve the manuscript quality, the authors are suggested to address the following questions:

We would like to thank the reviewer for their understanding of the revised manuscript.

1. The interconnection of d31 and d33 doesn't matter. For the d31 measurement, the authors may test the strain – electric field relationship, or indirectly deduce d31 from d33. For piezoelectric materials, d31 is also a critical parameter, the authors should at least provide an approximate value.

We thank the reviewer for their comment, however we disagree on the requirement for reporting the d31 in this particular study.

While the d31 is a critical parameter in piezoelectric materials, the d31 in PVDF-TrFE has been reported as lower than the d33, which already limits its utility as the primary electromechanical mechanism. Moreover, the submitted manuscript has demonstrated the polarization locking specifically in the z direction, which would, from a theoretical perspective, reduce the polarization in the 1 direction.

The interconnection between d33 and d31 following the direct piezoelectric effect is relevant, as the measured electrical displacement (per unit applied stress) in the 33 direction is reduced from the influences in the 31 direction *via* the Poisson's ratio, compliance and the Young's modulus. This is demonstrated clearly in Equation 1, as derived by K. Lefki and G. J. M. Dormans.¹

$$d_{33} = \frac{\partial D_3}{\partial T_3} = d_{31} + \frac{2d_{32}}{1-\nu} \left(\frac{\sigma_x}{Y} + \nu \frac{\sigma_y}{Y} \right) + \frac{d_{33}}{1-\nu} \left(\frac{\sigma_x}{Y} + \nu \frac{\sigma_y}{Y} \right)$$

Here, D_3 is the induced electrical displacement, T_3 is the applied stress, s_{31}^E , s_{11}^E and s_{12}^E are the mechanical compliances in their respective directions at constant electric field ($E = 0$), ν is the Poisson's ratio and Y is the Young's modulus.

Thus, in the presence of a large enough magnitude of d_{31} , the measured d_{33} would be reduced. This was not observed in our experiments, as the d_{33} was in fact higher than the literature values for electrically poled PVDF-TrFE.²

Building an accurate strain-based d_{31} measurement system (converse piezoelectric effect) is not practical, especially considering we are estimating $d_{31} \ll 10 \text{ pm V}^{-1}$ (from Equation 1), as the electric field would be applied in the z direction and the strain must be monitored in the x direction (Figure R1a). Importantly, the film cannot be clamped, due to the requirement for constant stress as shown in Equation 2, whereby the material will also exhibit an expansion in the z direction and affect the measurement.¹

$$d_{31} = \frac{\partial D_3}{\partial E_1} = \frac{\partial P_3}{\partial E_1} \quad \text{Eq. 2}$$

For an unclamped flexible film (Figure R1b), such miniscule changes in the x direction would (1) be difficult to detect, and (2) the measurement would be hindered by the low mechanical flexibility of the film, thus reducing the accuracy and reliability of the measured values.

Figure R1: Schematic demonstrating the d_{31} measurement through the converse piezoelectric effect for **a** a flat, non-flexible film and **b** a flexible film.

Notably, the d_{33} values are often favored over the d_{31} values for fluoropolymer films,³⁻⁸ unless the tensile elongation is the primary energy conversion mechanism, which is not the case in this work.

2. There are some other papers (See “High-performance micromachined unimorph actuators based on electrostrictive poly (vinylidene fluoride–trifluoroethylene) copolymer” and “Insect-scale fast moving and ultrarobust soft robot”) introducing the PVDF or PVDF-TrFE actuators. For an actuator, the authors can coat some other elastic material to the PVDF-TrFE film or fabricate an actuator by laminating method.

We thank the reviewer for their comment on fabricating cantilever actuators. Cantilever actuators currently form one application for piezo- and ferro-electric materials. This manuscript is not attempting to develop new actuators. This manuscript demonstrates a fundamental material property and consequently we do not want to limit the scope of the applicability of this polarization-locked system. The manuscript presents the in-depth analysis of the electromechanical energy conversion parameters as the fundamental quantity. These energy conversion parameters are subsequently applicable to cantilever actuators, as well as film actuators and stress sensors (conventional applications). The manuscript further describes the envisaged utility of such a polarization locked system to emerging applications in the main text, with the relevant text added below for convenience.

“The highly valorized commercial applications for piezoelectric materials, including precision motorized stages and inkjet printheads, utilize the converse piezoelectric effect,⁷ converting an applied electric field to discrete mechanical outputs.⁶ In contrast, emerging applications which utilize the direct piezoelectric effect⁷ to convert mechanical to electrical energy, the electrical poling process is a roadblock to commercialization, requiring a higher energy input than can be harvested in the device lifespan. These emergent applications, including energy harvesting,^{3,8} robotic interfaces,⁹ piezocatalysis⁶ and piezophotonics,⁴⁸ require revisiting of the electrical poling process and examination of the pathways for inducing polarization without high input energies.^{1,2}”

and

“Tuning surface terminations on MXenes and other 2D materials could afford enhanced electrostatic interactions leading to further improvements in piezoelectric outputs in fluoropolymers. Leveraging this new understanding of nanoscale phenomena at the interface of a fluoropolymer and a 2D sheet now opens up a plethora of research opportunities to design piezoelectric composites with broad applicability, including in wearable energy harvesting,⁸ piezo-catalysis,⁶ piezo-photonics,⁴⁸ and anisotropic sensors.⁴⁸ Coupled to the elegant and versatile fabrication methods, our sustainable system can enable bespoke device design in emerging fields for robotic interfaces,⁹ biomedical implants,⁹ and direct-on-organ printed electronics.³⁷”

This polarization-locked system unlocks further possibilities in ferroelectric applications, for example, random access memory (RAM),⁹ data storage¹⁰ and logic devices,¹¹ whereby the polarization direction and polarization magnitude can be tailored through electrostatic interactions between a non-piezoelectric additive and the fluoropolymer, without the requirement of electrically poling to a particular state prior to integration.

3. The demonstration of three-phase PVDF-TrFe copolymer with dihedral angle and trans/gauche-state is nice. The authors have demonstrated the α and γ phases are adjacent to $\text{Ti}_3\text{C}_2\text{T}_x$ where β phase is the main conformation in the bulk composite. Is there a quasi-morphotropic boundary or will there be a micro/elemental distortion in the bulk composites? Is there a change in the net polarization or Curie Point for the nanosheet based on the distribution and phase composition or fraction?

We thank the reviewer for their positive comments regarding the updated manuscript. The Molecular Dynamics simulations have demonstrated that the polarization vector undergoes a gradual reorientation from a randomized directionality to a locked directionality as a function of separation from the $\text{Ti}_3\text{C}_2\text{T}_x$ nanosheet. From these experiments, the manuscript has shown the highly attractive force between the nanosheet and the polymer chains. This attractive force enables the polymer chains directly adjacent to the nanosheet ($<35 \text{ \AA}$) to change to the phases with reduced piezoelectric activity (Figure 2c in the main text, shown in Figure R2c for convenience) due a depolarizing electrostatic field at such short length scales;¹² however, the polymer chains at separations of $\geq 35 \text{ \AA}$ from the $\text{Ti}_3\text{C}_2\text{T}_x$ nanosheet will only undergo spatial reorientation and thus the polarization vector is preferentially aligned. Without undertaking extensive additional high resolution transmission electron microscopy¹³ and synchrotron x-ray diffractometry experiments,¹⁴ deducing the behavior at the boundary between the α/γ phase ($<35 \text{ \AA}$ from the $\text{Ti}_3\text{C}_2\text{T}_x$ nanosheet) and the β phase ($\geq 35 \text{ \AA}$ from the $\text{Ti}_3\text{C}_2\text{T}_x$ nanosheet) is a non-trivial task.

Figure R2: **a,b** The top view (left column) and side view (right column) snapshots at $t = 0, 0.5, 1$ and 1.5 ns, with the resultant dipole moment vectors (right column) of the PVDF-TrFE copolymer film (blue arrow) and the **a** $Ti_3C_2T_x$ substrate (red arrow) or **b** graphene substrate. The annotated s and θ in **a** represent the separation of the fluoropolymer from the substrate and the PVDF-TrFE co-polymer dipole angle relative to the substrate, respectively. **c,d** Probability distributions of the angle (θ) between the dipole moment vector of the PVDF-TrFE co-polymer film layers and the **c** $Ti_3C_2T_x$ or **d** graphene substrates as a function of separation (s), as calculated from the equilibrated region of the obtained MD simulation.

Based on the prior experiments performed on fluoropolymer composites with non-piezoelectric additives (single-walled carbon nanotubes in PVDF-TrFE), the observed mechanism is the templating of the orientation of the β phase PVDF-TrFE.¹⁵

Regarding the second question on the changes in polarization or the Curie temperature of the $\text{Ti}_3\text{C}_2\text{T}_x$ nanosheets, neither of these are applicable. $\text{Ti}_3\text{C}_2\text{T}_x$ nanosheets do not exhibit piezoelectric properties due to the out-of-plane symmetry in the structure, which has been observed for the $\text{P63}/\text{mmc}$ point group.^{16,17} We have experimentally confirmed the lack of out-of-plane piezoelectricity in the $\text{Ti}_3\text{C}_2\text{T}_x$ nanosheets by piezoresponse force microscopy, with the data presented in Figure S20 of the Supplementary Information and shown in Figure R3 for convenience. No response was observed in the phase and amplitude traces under an applied voltage (converse piezoelectric effect) for the $\text{Ti}_3\text{C}_2\text{T}_x$ nanosheet (tracked by the arrows), thus no out of plane polarization exists in the nanosheets.

Figure R3: DART-PFM maps of $\text{Ti}_3\text{C}_2\text{T}_x$ nanosheets adsorbed on a gold (Au)-coated silicon (Si) wafer. **a-c** The topography traces. **d-f** The piezoelectric phase traces. **g-i** The piezoelectric amplitude traces. **a, d, g** Field of view corresponding to 5 μm . Scale bar represents 1 μm . **b, e, h** Field of view corresponding to 2 μm . Scale bar represents 500 nm. **c, f, i** Field of view corresponding to 1 μm . Scale bar represents 250 nm. The arrow in each panel points to the location of the same $\text{Ti}_3\text{C}_2\text{T}_x$ nanosheet.

Reviewer #3 (Remarks to the Author):

The authors have addressed all my concerns and questions. The improved version of the manuscripts looks pretty solid. I don't have any additional comment. The paper can be accepted.

We thank the reviewer for their detailed understanding of the manuscript and are pleased to hear of the positive outcome.

References

1. Lefki, K. & Dormans, G. J. M. Measurement of piezoelectric coefficients of ferroelectric thin films. *J. Appl. Phys.* **76**, 1764-1767 (1994).
2. Liu, Y. *et al.* Ferroelectric polymers exhibiting behaviour reminiscent of a morphotropic phase boundary. *Nature* **562**, 96-100 (2018).
3. Hafner, J. *et al.* Multi-scale characterisation of a ferroelectric polymer reveals the emergence of a morphological phase transition driven by temperature. *Nat. Commun.* **12**, 152 (2021).
4. Qiu, C. *et al.* Transparent ferroelectric crystals with ultrahigh piezoelectricity. *Nature*, **577**, 350-354 (2020).
5. Liu, Y. *et al.* Ferroelectric polymers exhibiting behaviour reminiscent of a morphotropic phase boundary. *Nature* **562**, 96-100 (2018).
6. Katsouras, I. *et al.* The negative piezoelectric effect of the ferroelectric polymer poly(vinylidene fluoride). *Nat. Mater.* **15**, 78-84 (2016).
7. Huang, Y. *et al.* Enhanced piezoelectricity from highly polarizable oriented amorphous fractions in biaxially oriented poly(vinylidene fluoride) with pure β crystals. *Nat. Commun.* **12**, 675 (2021).
8. You, L. *et al.* Origin of giant negative piezoelectricity in a layered van der Waals ferroelectric. *Sci. Adv.* **5**, eaav3780 (2019).
9. Lee, D. *et al.* Multilevel Data Storage Memory Using Deterministic Polarization Control. *Adv. Mater.* **24**, 402-406 (2012).
10. Aoki, T., Hiranaga, Y. & Cho, Y. High-density ferroelectric recording using a hard disk drive-type data storage system. *J. Appl. Phys.* **119**, 184101 (2016).
11. Lukyanchuk, I. *et al.* Ferroelectric multiple-valued logic units. *Ferroelectrics* **543**, 213-221 (2019).
12. Junquera, J. & Ghosez, P. Critical thickness for ferroelectricity in perovskite ultrathin films. *Nature* **422**, 506-509 (2003).
13. Kiss, Á. K., Rauch, E. F., Pécz, B., Szívós, J. & Lábár, J. L. A Tool for Local Thickness Determination and Grain Boundary Characterization by CTEM and HRTEM Techniques. *Microsc. Microanal.* **21**, 422-435 (2015).
14. McDonald, S. A. *et al.* Non-destructive mapping of grain orientations in 3D by laboratory X-ray microscopy. *Sci. Rep.* **5**, 14665 (2015).
15. Shepelin, N. A., *et al.* Printed recyclable and self-poled polymer piezoelectric generators through single-walled carbon nanotube templating. *Energy Environ. Sci.* **13**, 868-883 (2020).

16. Wang, H. W., Naguib, M., Page, K., Wesolowski, D. J. & Gogotsi, Y. Resolving the Structure of $\text{Ti}_3\text{C}_2\text{T}_x$ MXenes through Multilevel Structural Modeling of the Atomic Pair Distribution Function. *Chem. Mater.* **28**, 349–359 (2016).
17. Zou, W. N., Tang, C. X. & Pan, E. Symmetry types of the piezoelectric tensor and their identification. *Proc. R. Soc. A* **469**, 20120755 (2013).

REVIEWERS' COMMENTS:

Reviewer #2 (Remarks to the Author):

In the revised manuscript, the authors have addressed all the concerns and questions of the reviewer. The revised manuscript looks solid. The reviewer has no more comments or questions. The manuscript can be accepted.

DETAILED RESPONSE TO REVIEWERS

The authors would like to thank the editors and reviewers for their time, expertise, and dedication in reviewing our manuscript. Please find below a detailed point-by-point response to the reviewers' comments and recommendations.

Reviewer #2 (Remarks to the Author):

In the revised manuscript, the authors have addressed all the concerns and questions of the reviewer. The revised manuscript looks solid. The reviewer has no more comments or questions. The manuscript can be accepted.

We thank the reviewer for their feedback and are pleased with the recommendation for acceptance. We greatly appreciate the time and effort taken by the reviewers to evaluate our manuscript and provide valuable feedback in order to strengthen it, especially over this challenging time.